## [Transparent Peer Review file · Nature Communications]

Lactate derived from macrophages drives skin dermal fibroblasts phenotypic remodeling via MCT1-primed histone H3 lysine 23 lactylation in hypertrophic scar

Corresponding Author: Professor Dahai Hu

Version 0:

Reviewer comments:

Reviewer #1

(Remarks to the Author)

In this manuscript, the authors investigated the critical role of macrophage-derived lactate in the progression of hypertrophic scarring (HS), offering valuable insights into its pathogenesis. Initially, by comparing normal and HS human samples, the authors identified a significant metabolic shift from oxidative phosphorylation in normal tissue to glycolysis in HS, marked by notable lactate accumulation; subsequently, they demonstrated that macrophages, rather than fibroblasts or vascular endothelial cells, are the primary lactate producers in response to increased mechanical stiffness of the substrate extracellular matrix (ECM) in vitro. Mechanistically, the authors propose novel lactate—monocarboxylate transporter (MCT1)—histone H3 lysine 23 lactylation (H3K23la)—Hairy-related family bHLH transcription factor with YRPW motif 2 (HEY2)/collagen type XI (COL11)—Yes1 associated transcriptional regulator (YAP1)/Smad family member 2 (SMAD2) pathway to drive fibrotic processes in HS, supported by evidence that pharmacological inhibition of this cascade effectively reduces fibroblast activation and collagen deposition. Furthermore, utilizing a splinted excision wound model in mice with the C57BL/6 background, the authors found that sodium lactate delayed wound reepithelialization while disrupting Mct1 signaling pathways accelerated wound closure. The manuscript is well-crafted with minimal typographical errors and provides sufficiently detailed methods to ensure experimental reproducibility. However, several specific concerns have been identified during the review process that, in my opinion, should be addressed prior to publication.

Concerns with the manuscript:

1. It is unclear why gene expression data were presented by a heatmap (e.g., Figs. 1B, 3F, and 3K). While heatmaps with color gradients effectively summarize large datasets and may facilitate clustering to uncover gene-sample relationships, their application seems less advantageous here, given the limited comparison of only a few genes across two groups; furthermore, they fail to convey detailed information on data distribution (e.g., standard deviation) or statistical significance.
2. Some immunofluorescent staining figures (e.g., Figs. 1E, 1F, 6A, 6B, 6E, and 8D, as well as Supplementary Figs. 1B, 2A, 2C, 3D, 4D, 5D, 5E, 6D, 6E, and 9A) are presented in low quality, which hinders a thorough assessment of the data.
3. Transwell assays (e.g., Supplementary Figs. 1D, 2B, 2D, 3E, 4E, and 6E) should be quantified to enhance reliability, as it is widely recognized that single representative photographs may introduce artificial effects, particularly in transwell assays where uneven cell distribution is evident, as seen in the second panel of Supplementary Fig. 2B.
4. Supplementary Fig. 1E is cited in the main text before Supplementary Fig. 1B-D.
5. The authors compellingly demonstrated that a pathologically (high) stiff ECM enhances lactate production by macrophages, a key finding in this study. Since a high-stiff ECM is a defining characteristic of HS, reflecting the altered biomechanical and biochemical environment that drives excessive scar formation, it raises the question of whether HS formation, to some extent, precedes and triggers lactate production, potentially exacerbating the condition. The critical cause-and-effect relationship between HS initiation/establishment and lactate secretion should be demonstrated convincingly before moving to the other studies in this manuscript. Moreover, if HS onset precedes lactate release, it is critical to investigate whether and how the proposed strategy can effectively reverse this pathological process, which is essential for clinical applications in the real world.

6. Lines 236-237, 'We selected si-MCT1-1 or subsequent experiments which exhibited the highest knockdown efficiency.' Firstly, the study includes only two groups with the knockdown procedure, indicating a need for careful language revision. (A similar concern is also raised regarding the statements related to si-HEY2-1 and si-COL11A1-2.) Secondly, the relevant data (Supplementary Fig. 3A, second panel) do not support the statement, as no statistical differences are observed between si-MCT1-1 and si-MCT1-2 according to the provided source data.
7. Type XI collagen is a heterotrimer, comprising three distinct polypeptide chains—alpha1(XI), alpha2(XI), and alpha3(XI)—encoded by the genes Col11A1, Col11A2, and Col2A1, respectively. It is noteworthy that only Col11A1 appears to be affected, which merits a more thorough discussion to explore the underlying mechanisms and implications.
8. The quality of Fig. 7D and 8A is insufficient for proper assessment. It needs improvement to enable a thorough evaluation.
9. In Fig. 8E, the enlarged section: The notes of amino acid interactions are blurry. A similar illustration to Fig. 8F should be considered.
10. The splinted excisional wound model in mice used in the current study is a widely recognized tool for studying wound healing, as it effectively simulates crucial processes like re-epithelialization and granulation tissue formation that occur in humans. However, it falls short of being a standard model for HS. A compelling 2022 systematic review of HS animal models (PMID: 35743999) concluded that no murine model, including those utilizing splinted wounds, can truly replicate the intricacies of human hypertrophic scars. This limitation arises from fundamental differences in skin anatomy and healing mechanisms between species. It is essential to acknowledge these differences and consider using more suitable models for the current research.
11. Delayed wound closure is not a defining characteristic of HS. Instead, its occurrence is usually influenced by factors such as infection, inadequate blood supply, or underlying conditions like diabetes. While delayed healing can be seen as a general indicator of wound healing and may lead to chronic wounds (e.g., ulcers), it does not specifically lead to HS. However, it can create conditions that predispose wounds to HS, especially in situations involving prolonged inflammation or mechanical tension. In contrast, the formation of HS is more directly related to the quality of healing, particularly excessive deposition of ECM, rather than the speed of wound closure. For instance, HS can develop in wounds that heal quickly if they are accompanied by exaggerated inflammatory and fibrotic responses. Therefore, it is important to carefully reconsider the interpretation of animal study data in light of these insights.
12. Lines 400-402, 'According to the specific structure of the skin shown by HE staining, collagen arrangement was smoother and more orderly ...' Firstly, the description of 'the specific structure' requires more detail to enhance clarity and understanding. Secondly, incorporating established methods to quantitatively assess the smoothness and orderliness of collagen fibers would strengthen the supporting evidence for the statement.
13. The ratio of collagen I to collagen III undergoes a dynamic shift during the wound healing process, reflecting the stages of tissue repair and remodeling. Thus, the noted reduction in the collagen I/III ratio in Figs. 9 and 10 may be just a secondary effect of delayed healing, which should not be overrated.
14. Line 475: The reference to Zhang et al. should be provided.

Reviewer #2

(Remarks to the Author)

In the manuscript titled "Lactate derived from macrophages drives skin dermal fibroblasts phenotypic remodeling via MCT1-primed histone H3 lysine 23 lactylation in hypertrophic scar," the authors investigate how matrix stiffness-activated macrophages produce excess lactate, which is shuttled into dermal fibroblasts via MCT1. This influx promotes site-specific histone H3K23 lactylation (H3K23la), leading to upregulation of profibrotic genes HEY2 and COL11A1. Through CUT&Tag, RNA-seq, molecular simulations, and both genetic (fibroblast-specific Mct1 knockout) and pharmacological (AZD3965) interventions in murine wound models, they demonstrate that disrupting the MCT1-H3K23la-HEY2/COL11A1 feedback loop attenuates hypertrophic scar formation. The presented findings are new and supported by the experimental data. The reviewer has several comments listed below.

- 1) The immunofluorescence images in Figure 1F show MCT1 colocalization with α -SMA-positive myofibroblasts, but no quantitative colocalization analysis (e.g., Pearson's coefficient) is provided. Please include such metrics to substantiate the claim.
- 2) The rationale for using 10 ng/mL TGF- β is not explained. Cite relevant literature or provide preliminary data to justify this concentration.
- 3) The manuscript proposes that lactate enters fibroblasts via MCT1 to induce H3K23la and activate HEY2 and COL11A1 transcription, but lacks details on the dynamic regulation of this process. For example, the dose-response relationship between lactate concentration and H3K23la levels has not been fully characterized.
- 4) The authors state that 2 kPa and 50 kPa hydrogels were used to model physiological and pathological matrix stiffness, but no supporting references or experimental stiffness measurements are provided. Please cite precedent studies or supply experiment data.

Reviewer #3

(Remarks to the Author)

In their manuscript, entitled “Lactate derived from macrophages drives skin dermal fibroblasts’ phenotypic remodeling via MCT1-primed histone H3 lysine 23 lactylation in hypertrophic scar,” Yuan et al. investigate increased lactate production in the hypertrophic skin and its influence on fibroblast function in fibrosis and wound healing. The authors further determined that MCT1 functions to transport lactate into fibroblasts, where it is utilized for histone lactylation. This, in turn, activates the transcription of profibrotic genes HEY2 and COL1A1, which activate YAP1 expression and aggravate lactate accumulation, respectively. Finally, the authors test the relevance of MCT1 expression in skin fibroblasts using a wound healing model, employing genetic depletion and compound blocking.

This reviewer is deeply concerned about the novelty of the findings, the technical quality of the presented data, and the technical quality of the manuscript. The role of MCT1 in fibrosis and keloid fibroblasts has been known for some time (Jinhua Feng et al., 2024; David R. Ziehr et al., 2024 preprint; Kyounghee Min, 2023; Jing-Jing Gu 2025, and many others), as well as the role of lactylation in fibrosis (Huanwei Liang et al., 2025 review). Therefore, the novelty of the study results is limited, if any.

The presented manuscript is generally poorly written, contains numerous grammatical errors, and references data that are not present in the figures. Additionally, it includes too many figures that do not contribute to the study’s quality.

Major concerns:

1. The majority of the paper utilizes an in vitro system to investigate a complex and multifaceted disease that affects complex tissue, while only selected experiments employ tissue samples. It is unclear why these shifts in experimental systems are made, as they are by no means beneficial to the study.
2. In the first figure, authors proceed directly to measuring lactate levels in the hypertrophic skin, providing little justification for why they chose to present precisely this metabolite and not assess the overall metabolomic profile of hypertrophic versus normal skin. The authors should provide an extended metabolomic profile of hypertrophic versus normal skin, allowing for a comparison of the presence of different metabolites in the skin.
3. If the authors want to discuss shifts between energy-supplying pathways, they should demonstrate that the actual transition from one to another occurs. What is the relative expression of genes involved in oxidative phosphorylation? Is glycolysis indeed the dominating pathway used for energy generation in the hypertrophic skin?
4. The quality of IF is very poor, and it’s difficult to see individual cells. Higher quality imaging should be provided.
5. There is a discrepancy in the relative protein expression measured by Western blot, IHC, and IF in Figure 1. The authors should explain why.
6. The authors talk about data that is not present in the figure: “ and we did not observe noticeable differences of MCT1 intensity in vascular endothelial cells between NS and HS (Figure 1F)” – no markers of vascular endothelial cells has been used in the whole figure, and in particular not the referred Figure 1F.
7. In Figure 2, the authors use an in vitro system to investigate the cellular origin of lactate, which is detected in hypertrophic skin tissue. This approach is reductionistic, as it does not aim to identify which of the various cell types in the hypertrophic skin is the primary producer of lactate. Instead, they select specific cells to probe for lactate production. The authors should either use patient material (as in Figure 1) or the developed mouse model to answer this question.
8. The molecular interaction modeling exercise doesn’t contribute to the main message of the study.

Reviewer #4

(Remarks to the Author)

In the study titled “Lactate derived from macrophages drives skin dermal fibroblasts phenotypic remodeling via MCT1-primed histone H3 lysine 23 lactylation in hypertrophic scar,” Hu et al. delineated the role of macrophage-derived lactate in hypertrophic scar (HS) formation. They identified that macrophage-derived lactate serves as a critical mediator of fibroblast phenotypic remodeling through monocarboxylate transporter 1 (MCT1)-mediated histone H3 lysine 23 lactylation (H3K23la) in HS. The authors observed that fibroblast-specific deletion of Mct1 or pharmacological inhibition of Mct1 in mice reduced collagen deposition, accelerated wound healing, and attenuated scar formation.

The study is highly interesting, well designed, and clearly presented. Nevertheless, several minor issues need to be addressed:

1. Multiple immunofluorescence images in the manuscript are of relatively low resolution (e.g., Figure 1E and Figure S6). Higher-resolution images should be provided.
2. In Figure 9C and Figure 10C, the blue coloration of the Masson’s trichrome staining is uneven, suggesting possible overstaining. Higher-resolution Masson’s staining images are required.
3. The authors demonstrated in vitro that macrophages are the primary source of lactate accumulation in the scar metabolic microenvironment. Given the substantial differences between in vitro and in vivo conditions, in vivo validation of macrophage-mediated lactate accumulation in the hypertrophic scar microenvironment would further strengthen the credibility of this conclusion.

4. The authors identified an interaction between COL11A1 and MCT1. However, it remains unclear whether additional proteins are involved in this interaction. Further experimental validation or detailed discussion in the Discussion section is warranted.

5. The authors observed elevated lactate levels and increased MCT1 expression in HS tissues. What is the relationship between lactate levels, MCT1 expression, and HS disease progression?

6. In Figure S6C, nonspecific bands are present for COL11A1, and the two bands in the si-NC group are difficult to distinguish. A clearer Western blot image should be provided.

Version 1:

Reviewer comments:

Reviewer #1

(Remarks to the Author)

All reviewer comments have been adequately addressed in the revised manuscript, supported by the inclusion of additional data. The revisions are thorough and acceptable for publication.

Reviewer #2

(Remarks to the Author)

the revision has addressed the reviewer's comments. there are no further comments

Reviewer #3

(Remarks to the Author)

It's appreciated that the authors have performed additional experiments in response to the previous review. These efforts have improved the manuscript and clarified several aspects of the study. Nonetheless, several important points remain outstanding, and further clarification and supporting evidence are needed to validate the study's conclusions fully:

Regarding the novelty of the findings

The authors have provided a detailed explanation of their view on the novelty of the study and have clarified how their findings integrate previously known concepts into a more specific mechanistic framework. While these additions and contextual revisions are appreciated, this reviewer remains unconvinced that the presented work constitutes a substantial conceptual advance. The roles of MCT1-mediated lactate transport, macrophage-derived lactate signaling, and histone lactylation in fibroblast activation have been previously established across several fibrotic contexts, including keloid and dermal fibrosis. The current study extends these findings to hypertrophic scar and delineates their interconnection in this specific model, but this represents more of a contextual refinement than a discovery of a fundamentally new mechanism. While the addition of new molecular players (e.g., H3K231a, HEY2, and COL11A1) provides valuable mechanistic depth, the overarching concept of lactate-mediated metabolic–epigenetic crosstalk driving fibrosis remains consistent with prior reports. Thus, the study represents a meaningful refinement and contextualization of established mechanisms rather than the identification of a fundamentally new pathway.

Original Comment 1:

Using in vitro model systems to dissect mechanistic details is acceptable when grounded in initial observations derived from the native tissue context. However, in this study, the focus on fibroblasts, macrophages, and endothelial cells appears selective rather than empirically established. A more systematic analysis that includes other key skin-resident cell types—such as adipocytes, mast cells, and additional immune populations—would have been essential to accurately identify the principal contributors to the observed interactions. Moreover, the use of cell lines for in vitro experiments represents a significant limitation, as these models often diverge from primary cells and therefore cannot faithfully reproduce the complexity of tissue-level signaling and cell–cell communication. This methodological choice reduces the physiological relevance of the mechanistic findings and may limit the generalizability of the conclusions to in vivo or clinical settings.

Original Comment 2:

While this reviewer acknowledges the authors' effort in performing a non-targeted metabolomic analysis, the rationale for selecting lactate as the central metabolite of interest still remains insufficiently justified. Specifically, it is unclear whether lactate was directly detected and quantified in the presented metabolomic dataset, and if so, whether it represents the most significantly upregulated metabolite in hypertrophic scar (HS) compared with normal skin. If lactate is not the top metabolite showing increased abundance, the authors should clarify why it was prioritized over other metabolites with potentially stronger differential expression. Furthermore, identifying the metabolite(s) with the highest upregulation and discussing their potential relevance would provide critical context, ensuring that the mechanistic focus on lactate is data-driven rather than selectively interpretive.

Original Comment 3:

The addition of transcript-level measurements for selected TCA cycle enzymes is appreciated; however, this approach alone

does not constitute definitive evidence of a metabolic shift from oxidative phosphorylation (OXPHOS) to glycolysis in hypertrophic scar (HS) tissue. Given that the authors have already provided Western blot data for glycolysis-related proteins, equivalent protein-level validation for the selected OXPHOS-associated genes would be necessary for methodological consistency and to strengthen this comparison.

Furthermore, the added metabolomic analysis requires deeper exploration. The authors should provide a more comprehensive overview of the metabolites involved in OXPHOS and glycolysis, demonstrating whether OXPHOS intermediates are indeed reduced in HS and whether glycolytic intermediates are correspondingly elevated. Highlighting only a single metabolite (citric acid) from the TCA cycle does not sufficiently support the claim of an energy pathway shift, particularly since other relevant pathways, e.g., fatty acid oxidation, have not been considered.

Finally, expanding the Seahorse data presented in Figure 2 to include an OCR/ECAR ratio would provide a functional readout of the relative balance between oxidative and glycolytic metabolism, thereby offering further support for the proposed shift toward glycolysis.

Original Comment 7:

While this reviewer acknowledges the substantial effort undertaken to isolate primary cells from patient skin samples and assess their contribution to lactate production, the rationale for excluding other skin-resident cell types remains unclear. Additional cell populations—such as adipocytes, mast cells, and other immune subsets—could plausibly contribute to the altered metabolic landscape of hypertrophic scars. The authors should either extend their analysis to include these dominant skin cell types or provide a clear and experimentally grounded justification for their exclusion. Addressing this point would ensure that the presented conclusions regarding the cellular origin of lactate are comprehensive and not selectively derived.

Original Comment 8:

Without experimental verification, the molecular modeling remains speculative. The authors should indicate whether the model's predictions can be experimentally tested. Validation through targeted mutagenesis or binding assays would significantly enhance the credibility of these computational results.

Reviewer #4

(Remarks to the Author)

The authors have been very responsive to both my and the comments of the other reviewers. The manuscript has been further improved and I have no additional comments.

Version 2:

Reviewer comments:

Reviewer #3

(Remarks to the Author)

The manuscript has been substantially improved. Although this reviewer still has some reservations regarding the rationale and justification for selecting lactate as the key metabolite, the overall quality of the work warrants publication in Nature Communications.

Response to Reviewers

Dear Editors and Reviewers,

Thank you for your letter and for the reviewers' comments concerning our manuscript entitled "Lactate derived from macrophages drives skin dermal fibroblasts phenotypic remodeling via MCT1-primed histone H3 lysine 23 lactylation in hypertrophic scar" (Manuscript Number: NCOMMS-25-39200). Those comments are all valuable and very helpful for revising and improving our manuscript, as well as the important guiding significance to our researches. We have studied comments carefully and have made corrections which we hope meet with approval. Revised portions are marked in yellow in the revised manuscript. The main corrections in the paper and the responses to the reviewer's comments are as following:

Responses to the reviewer's comments:

Reviewer #1:

In this manuscript, the authors investigated the critical role of macrophage-derived lactate in the progression of hypertrophic scarring (HS), offering valuable insights into its pathogenesis. Initially, by comparing normal and HS human samples, the authors identified a significant metabolic shift from oxidative phosphorylation in normal tissue to glycolysis in HS, marked by notable lactate accumulation; subsequently, they demonstrated that macrophages, rather than fibroblasts or vascular endothelial cells, are the primary lactate producers in response to increased mechanical stiffness of the substrate extracellular matrix (ECM) *in vitro*. Mechanistically, the authors propose novel lactate—monocarboxylate transporter (MCT1)—histone H3 lysine 23 lactylation (H3K23la)—Hairy-related family bHLH transcription factor with YRPW motif 2 (HEY2)/collagen type XI (COL11)—Yes1 associated transcriptional regulator (YAP1)/Smad family member 2 (SMAD2) pathway to drive fibrotic processes in HS, supported by evidence that pharmacological inhibition of this cascade effectively reduces fibroblast activation and collagen deposition. Furthermore, utilizing a splinted excision wound model in mice with the C57BL/6 background, the authors found that sodium lactate delayed wound reepithelialization while disrupting Mct1 signaling pathways accelerated wound closure. The manuscript is well-crafted with minimal typographical errors and provides sufficiently detailed methods to ensure experimental reproducibility. However, several specific concerns have been identified during the review process that, in my opinion, should be addressed prior to publication.

Concerns with the manuscript:

Comment #1: It is unclear why gene expression data were presented by a heatmap (e.g., Figs. 1B, 3F, and 3K). While heatmaps with color gradients effectively summarize large datasets and may facilitate clustering to uncover gene-sample relationships, their application seems less advantageous here, given the limited comparison of only a few genes across two groups; furthermore, they fail to convey detailed information on data distribution (e.g., standard deviation) or statistical significance.

Reply:

Thanks for your suggestions. We have carefully considered your valuable comment regarding the use of heatmaps. We fully agree with your point that heatmaps are not the most effective way to present data when comparing a limited number of genes between only two groups. Following your recommendation, we have revised the figures and have replaced the heatmaps in Figure 1B, 3F, and 3K with bar graphs in the revised manuscript. The new figures clearly display the mean and standard deviation of the gene expression data and, most importantly, include statistical significance to provide a more comprehensive and accurate representation of our findings. Thank you again for your guidance.

Revised figures:

Figure 1B

Figure 3F

Figure 3K

Comment #2: Some immunofluorescent staining figures (e.g., Figs. 1E, 1F, 6A, 6B, 6E, and 8D, as well as Supplementary Figs. 1B, 2A, 2C, 3D, 4D, 5D, 5E, 6D, 6E, and 9A) are presented in low quality, which hinders a thorough assessment of the data.

Reply:

Thank you for your valuable feedback and the time you have dedicated to reviewing our manuscript. We sincerely apologize for the low quality of the immunofluorescent images in the original submission. We agree that high-quality images are crucial for the interpretation of

the data presented. In response to your comment, we have carefully revised all the figures you mentioned. Specifically, we have replaced the original images in Figures 1E, 1F, 6A, 6B, 6E, and 8D, as well as in Supplementary Figures 2B, 4A, 4C, 6D, 7D, 8B, 8C, 9F, 9G, and 11D, with new, high-resolution versions. These new images provide much greater clarity and detail, which we believe will allow for a comprehensive and accurate assessment of our findings. As seen in the revised manuscript, these figures now clearly show the co-localization and expression levels of the stained markers. For example, in the updated Figure 1F, the specific upregulation of MCT1 in α -SMA⁺ myofibroblasts in hypertrophic scar tissue is now much more distinct. Similarly, the revised Figures 6A and 6B now provide a clearer visualization of the colocalization of HEY2 and COL11A1 with α -SMA in hypertrophic scar dermal myofibroblasts. We have ensured that all images in the revised manuscript meet the high-quality standards required for publication. We are confident that these changes have significantly strengthened our manuscript and have addressed your concerns. Once again, we thank you for your insightful feedback. We hope that the revised manuscript is now suitable for publication.

Revised figures:

Figure 1F

Figure 6A

Figure 6B

Comment #3: Transwell assays (e.g., Supplementary Figs. 1D, 2B, 2D, 3E, 4E, and 6E) should be quantified to enhance reliability, as it is widely recognized that single representative photographs may introduce artificial effects, particularly in transwell assays where uneven cell distribution is evident, as seen in the second panel of Supplementary Fig. 2B.

Reply:

Thanks for your suggestions. We appreciate your insightful comments regarding the Transwell assays. We agree that quantification is crucial for enhancing the reliability of these results and that relying solely on single representative photographs can indeed introduce potential artifacts, especially given the possibility of uneven cell distribution. For each experiment, we captured multiple images from different areas of each Transwell membrane. The number of migrated cells was then counted and the data are now presented as graphs, showing the mean and standard deviation from replicate experiments. We have now performed quantification for all Transwell assays presented in the manuscript (Figure 6G, 6H and Supplementary Figure 2E, 4B, 4D, 6F, 7F). Furthermore, we have specifically addressed your concern about the uneven cell distribution observed in the second panel of Supplementary Figure 4B. We have replaced the previous representative image with a new one that accurately reflects the results of the quantified data. Thank you again for your guidance.

Revised figure:

Figure 6G

Figure 6H

Figure S2E

Figure S4B

Figure S4D

Figure S6F

Figure S7F

Comment #4: Supplementary Fig. 1E is cited in the main text before Supplementary Fig. 1B-D.

Reply:

Thanks for your suggestions. We apologize for this error in the previous version of the manuscript. As suggested, we have revised the manuscript to ensure that all supplementary figures are cited in the correct order. The citation for Supplementary Figure 2B-D now precedes Supplementary Figure 2E in the main text. Thank you again for your guidance.

Revised figures:

Figure S2

Comment #5: The authors compellingly demonstrated that a pathologically (high) stiff ECM enhances lactate production by macrophages, a key finding in this study. Since a high-stiff ECM is a defining characteristic of HS, reflecting the altered biomechanical and biochemical

environment that drives excessive scar formation, it raises the question of whether HS formation, to some extent, precedes and triggers lactate production, potentially exacerbating the condition. The critical cause-and-effect relationship between HS initiation/establishment and lactate secretion should be demonstrated convincingly before moving to the other studies in this manuscript. Moreover, if HS onset precedes lactate release, it is critical to investigate whether and how the proposed strategy can effectively reverse this pathological process, which is essential for clinical applications in the real world.

Reply:

We sincerely thank the reviewer for this insightful and thought-provoking comment. This question touches upon the critical 'chicken-and-egg' dilemma regarding the temporal sequence of events in HS pathogenesis and is fundamental to understanding the therapeutic potential of our proposed strategy. We appreciate the opportunity to clarify our perspective on this crucial cause-and-effect relationship.

We concur with the reviewer that a stiff ECM is a hallmark of an established hypertrophic scar. However, we respectfully propose that the process is not a simple linear event where a fully formed, stiff HS matrix is a prerequisite for lactate production. Instead, we believe it is a self-amplifying positive feedback loop that begins much earlier in the HS formation cascade, with inflammation being the initial trigger (Younesi FS et al., 2024).

Our proposed sequence of events is as follows: **(1) Initiation by inflammation:** The process begins with the initial skin injury, which immediately triggers a robust inflammatory phase of wound healing. This phase is characterized by the infiltration of various immune cells, most notably macrophages, into the wound site. These macrophages are activated by damage-associated molecular patterns (DAMPs) and pro-inflammatory cytokines present in the early wound environment (Oscar A Peña et al., 2024). **(2) Early metabolic reprogramming of macrophages:** It is well-established that pro-inflammatory (M1-like) macrophages, which dominate the early phase of wound healing, undergo significant metabolic reprogramming (Yi-Kai Hong et al., 2023). They shift from oxidative phosphorylation towards aerobic glycolysis (the Warburg effect) (Zitong Wang et al., 2024; Sabine A Eming et al., 2023). A key consequence of this metabolic switch is the substantial production and secretion of lactate. This initial wave of lactate production occurs before the development of a pathologically stiff, fibrotic matrix. **(3) Lactate-driven fibroblast activation:** As our study demonstrates, this early, inflammation-driven lactate acts as a crucial signaling molecule for dermal fibroblasts. The influx of lactate into fibroblasts via MCT1, as shown in our paper, primes histone

lactylation (specifically H3K231a) and promotes their phenotypic transition into hyperactive, pro-fibrotic myofibroblasts. These activated myofibroblasts are the primary source of excessive collagen and other ECM component deposition. **(4) Establishment of a vicious positive feedback loop:** The excessive deposition of ECM by these myofibroblasts begins to increase the stiffness of the matrix. As we compellingly demonstrated in our *in vitro* experiments using PDMS hydrogels of varying stiffness (Figure 2), and as the reviewer kindly noted, this stiff mechanical microenvironment is a potent stimulus for macrophage lactate production. This creates a vicious, self-perpetuating cycle: Inflammation → Macrophage glycolysis → Lactate secretion → Fibroblast activation → ECM stiffening → Enhanced macrophage glycolysis → More lactate → Worsening fibrosis. This feedback loop explains how a normal healing process can be derailed into the pathological state of hypertrophic scar and why the condition persists.

Regarding the second comment, our therapeutic strategy, targeting the glycolysis-MCT1-H3K231a axis, is designed precisely to break this pathological feedback loop. By inhibiting MCT1 (AZD3965), we block the ability of fibroblasts to take up and respond to the macrophage-derived lactate. This intervention decouples the inflammatory and fibrotic processes. Even if the scarring process has been initiated and some level of matrix stiffening is present, inhibiting MCT1 would still dampen the pro-fibrotic response of fibroblasts, reduce their activation and collagen production, and prevent the further amplification of the cycle. Our *in vivo* studies (Figures 9, 10 and S11), where we applied either genetic deletion of *Mct1* in fibroblasts or pharmacological inhibition with AZD3965 in a full-thickness wound model, support this concept. In these experiments, the intervention was initiated after the wound was created, meaning the inflammatory and healing processes were already underway. The resulting accelerated wound healing and attenuated scar formation strongly suggest that disrupting this lactate-driven axis is effective at mitigating the fibrotic outcome, not just in a preventative setting but also during the active phase of scar formation. Therefore, this strategy holds the potential to effectively reverse or at least halt the progression of the pathological process, which is essential for real-world clinical applications.

To ensure this critical cause-and-effect relationship is more clearly articulated for our readers, we have revised the Discussion sections of our manuscript to explicitly detail this proposed temporal sequence and highlight the central role of the positive feedback loop in the establishment and progression of hypertrophic scar. Thank you again for your guidance.

Revised manuscript:

“Our findings in HS pathogenesis point to a self-amplifying "mechano-metabolic vicious cycle," rather than a simple linear causality between matrix stiffening and lactate production. We posit that initial wound healing processes, including inflammation and ECM deposition, stiffen the tissue microenvironment. This mechanical cue, as demonstrated in vitro and in vivo, triggers a metabolic shift in macrophages towards increased glycolysis and lactate secretion. Subsequently, fibroblast uptake of macrophage-derived lactate via MCT1 instigates H3K231a-dependent pathological activation and excessive ECM production. This not only intensifies fibrosis but also further stiffens the matrix, locking in a positive feedback loop that drives HS formation. Crucially, our work identifies this amplification loop as a novel therapeutic target. Pharmacological inhibition of MCT1 validated this concept, effectively accelerating wound closure and mitigating scar formation. This demonstrates that therapeutically interrupting this lactate-driven, mechano-metabolic crosstalk can steer the repair process away from pathological fibrosis, even after the initial injury. Though the potential of this strategy to reverse established, chronic scar remains a key area for future studies, as our current investigation centered on the proliferative phase of scarring.”

References:

- (1) Younesi, F. S., Miller, A. E., Barker, T. H., Rossi, F. M. V., & Hinz, B. (2024). Fibroblast and myofibroblast activation in normal tissue repair and fibrosis. *Nature reviews. Molecular cell biology*, 25(8), 617–638.
- (2) Peña, O. A., & Martin, P. (2024). Cellular and molecular mechanisms of skin wound healing. *Nature reviews. Molecular cell biology*, 25(8), 599–616.
- (3) Hong, Y. K., Chang, Y. H., Lin, Y. C., Chen, B., Guevara, B. E. K., & Hsu, C. K. (2023). Inflammation in Wound Healing and Pathological Scarring. *Advances in wound care*, 12(5), 288–300.
- (4) Wang, Z., Zhao, F., Xu, C., Zhang, Q., Ren, H., Huang, X., He, C., Ma, J., & Wang, Z. (2024). Metabolic reprogramming in skin wound healing. *Burns & trauma*, 12, tkad047.
- (5) Eming, S. A., Murray, P. J., & Pearce, E. J. (2021). Metabolic orchestration of the wound healing response. *Cell metabolism*, 33(9), 1726–1743.

Comment #6: Lines 236-237, ‘We selected si-MCT1-1 for subsequent experiments which exhibited the highest knockdown efficiency.’

Firstly, the study includes only two groups with the knockdown procedure, indicating a need for careful language revision. (A similar concern is also raised regarding the statements related to si-HEY2-1 and si-COL11A1-2.)

Secondly, the relevant data (Supplementary Fig. 3A, second panel) do not support the statement, as no statistical differences are observed between si-MCT1-1 and si-MCT1-2 according to the provided source data.

Reply:

Thanks for your suggestions. We have carefully considered your suggestions and have revised the manuscript accordingly to address the concerns you raised. Regarding your comments on lines 236-237, we acknowledge that our original phrasing and data presentation were not sufficiently robust. Firstly, we agree that with only two siRNA groups, the term "highest" was not the most appropriate. We have now expanded our study to include three different siRNA sequences for MCT1, as well as for P300, HEY2 and COL11A1, to allow for a more rigorous selection of the most effective siRNA. Furthermore, we have conducted additional RT-qPCR experiments to provide a more comprehensive and statistically sound validation of the knockdown efficiency for each siRNA. The new data, now presented in the revised Figure S6A, S6B, S7A, S7B and S9B-S9E, clearly demonstrates that si-MCT1-1 (and similarly si-P300-1, si-HEY2-1 and si-COL11A1-3) exhibits the most significant knockdown efficiency among the tested sequences. We are confident that these additions and the corresponding revisions to the text have addressed your concerns and strengthened this aspect of our study and we have revised the Result sections of our manuscript. We are grateful for your constructive criticism, which has significantly improved the quality and clarity of our manuscript. Thank you again for your guidance.

Revised figures:

Figure S6A-S6B

Figure S7A-S7B

Figure S9B-S9C

Figure S9D-S9E

Comment #7: Type XI collagen is a heterotrimer, comprising three distinct polypeptide chains—alpha1(XI), alpha2(XI), and alpha3(XI)—encoded by the genes Col11A1, Col11A2, and Col2A1, respectively. It is noteworthy that only Col11A1 appears to be affected, which merits a more thorough discussion to explore the underlying mechanisms and implications.

Reply:

Thanks for your suggestions. We appreciate your expertise and the thoughtful consideration you have given to our work. We concur that the specific upregulation of *COL11A1* without a corresponding increase in *COL11A2* and *COL2A1* is a significant finding that requires further elaboration. Our initial unbiased screening using CUT&Tag and RNA-seq analyses pointed specifically to an epigenetic and transcriptional upregulation of *COL11A1* in HS. Based on your feedback, we have further explored the literature to provide a more comprehensive discussion of the potential reasons for this specific observation.

There are several potential explanations for the specific upregulation of *COL11A1* in the context of hypertrophic scar: **(1) Distinct transcriptional regulation:** The genes encoding the three alpha chains of type XI collagen are located on different chromosomes and are subject to independent transcriptional control (Karl E Kadler et al., 2008). This allows for their expression to be differentially regulated in response to specific cellular signals and microenvironmental cues. Our study identifies a lactate-rich microenvironment as a key driver of the profibrotic phenotype in HS. We demonstrate that H3K231a, a direct consequence of elevated lactate, is enriched at the promoter of *COL11A1*, leading to its transcriptional activation. It is highly probable that the promoters of *COL11A2* and *COL2A1* do not possess the same sensitivity to this specific epigenetic modification, leading to the selective upregulation of *COL11A1*. **(2) Non-canonical roles of COL11A1 beyond the heterotrimer:** While classically viewed as a component of the type XI collagen heterotrimer, there is growing evidence that COL11A1 can have significant biological functions independent of its role in the trimer, particularly in pathological conditions like cancer (Sameera Nallanthighal et al., 2021; Yihui Wu et al., 2021). Several studies have highlighted that COL11A1 can be secreted as a monomer or homotrimer and can interact with the extracellular matrix and cell surface receptors to promote fibroblast activation, migration, and matrix deposition (Jiayu Zhang et al., 2023). Our findings support such a non-canonical role, where we show a novel interaction between COL11A1 and the lactate transporter MCT1. This interaction stabilizes MCT1 and enhances lactate influx, thereby creating a positive feedback loop that perpetuates the fibrotic state. This function is likely independent of the assembly of the full type XI collagen heterotrimer. **(3) COL11A1 as a Key Pro-fibrotic Mediator:** The upregulation of *COL11A1* has been specifically identified as a hallmark of various fibrotic conditions. For instance, it is a known biomarker for cancer-associated fibroblasts (CAFs) and is implicated in promoting a fibrotic tumor microenvironment that enhances tumor progression (Jiayu Zhang et al., 2023; Xuzhen Yan et al., 2025). Furthermore, its expression has been found to be elevated in keloids, a fibroproliferative disorder closely related to hypertrophic scar (Pingping Lin et al., 2022). This consistent association of *COL11A1* with fibrosis across different pathologies suggests that it may be a more central and primary driver

of the fibrotic process compared to the other two chains of type XI collagen. (4) **Alternative Chain Composition in Pathological ECM:** The composition of collagen heterotrimers can be altered in different tissues and under pathological conditions. It is possible that in the disorganized and rapidly remodeling ECM of a hypertrophic scar, the stoichiometric balance of collagen chain synthesis is disrupted. The pronounced upregulation of the $\alpha 1(XI)$ chain may lead to its incorporation into other collagen fibril types or the formation of atypical collagen structures that contribute to the pathological stiffness and density of the scar tissue. In summary, we propose that the specific upregulation of *COL11A1* in hypertrophic scars is not an anomaly but rather a key mechanistic event driven by the unique metabolic and epigenetic landscape of the scar microenvironment. The lactate-induced epigenetic activation of *COL11A1*, coupled with its potent, non-canonical pro-fibrotic functions, positions it as a critical driver in the pathogenesis of hypertrophic scar.

We have now incorporated this detailed explanation into the Discussion section of our revised manuscript to provide a clearer and more complete interpretation of our findings. Thank you again for your guidance.

Revised manuscript:

“The selective upregulation of COL11A1 in HS, independent of its heterotrimeric partners COL11A2 and COL2A1, is driven by metabolic-epigenetic crosstalk. The genes for each alpha chain are subject to independent transcriptional control. Specifically, the lactate-rich HS microenvironment promotes H3K231a enrichment at the *COL11A1* promoter, selectively activating its transcription. Furthermore, the COL11A1 protein exhibits a non-canonical function by directly interacting with the lactate transporter MCT1. Molecular modeling shows this interaction stabilizes MCT1 and enhances its lactate affinity, potentiating lactate transport. This establishes a pathogenic, self-reinforcing positive feedback loop (lactate-H3K231a-COL11A1-MCT1), which sustains the scar microenvironment.”

References:

- (1) Kadler, K. E., Hill, A., & Canty-Laird, E. G. (2008). Collagen fibrillogenesis: fibronectin, integrins, and minor collagens as organizers and nucleators. *Current opinion in cell biology*, 20(5), 495–501.
- (2) Nallanthighal, S., Heiserman, J. P., & Cheon, D. J. (2021). Collagen Type XI Alpha 1 (COL11A1): A Novel Biomarker and a Key Player in Cancer. *Cancers*, 13(5), 935.

(3) Wu, Y. H., Huang, Y. F., Chang, T. H., Chen, C. C., Wu, P. Y., Huang, S. C., & Chou, C. Y. (2021). COL11A1 activates cancer-associated fibroblasts by modulating TGF- β 3 through the NF- κ B/IGFBP2 axis in ovarian cancer cells. *Oncogene*, *40*(26), 4503–4519.

(4) Zhang, J., Lu, S., Lu, T., Han, D., Zhang, K., Gan, L., Wu, X., Li, Y., Zhao, X., Li, Z., Shen, Y., Hu, S., Yang, F., Wen, W., & Qin, W. (2023). Single-cell analysis reveals the COL11A1⁺ fibroblasts are cancer-specific fibroblasts that promote tumor progression. *Frontiers in pharmacology*, *14*, 1121586.

(5) Yan, X., Han, Q., Wu, W., Li, H., Zhang, W., Wang, Y., Chen, W., Yang, A., & You, H. (2025). ITGA8 deficiency in hepatic stellate cells attenuates CCl₄-Induced liver fibrosis via suppression of COL11A1. *Biochemical and biophysical research communications*, *756*, 151522.

(6) Lin, P., Zhang, G., Peng, R., Zhao, M., & Li, H. (2022). Increased expression of bone/cartilage-associated genes and core transcription factors in keloids by RNA sequencing. *Experimental dermatology*, *31*(10), 1586–1596.

Comment #8: The quality of Fig. 7D and 8A is insufficient for proper assessment. It needs improvement to enable a thorough evaluation.

Reply:

Thanks for your suggestions. We agree that their resolution was not optimal for a thorough evaluation, and we sincerely apologize for this oversight. We have now revised both figures to improve their clarity and detail in the revised manuscript. For Figure 7D, we would like to clarify that this figure was generated by the JASPAR database, a professional bioinformatics tool used for predicting transcription factor binding sites (Ieva Rauluseviciute et al., 2024). It illustrates the predicted binding regions of the HEY2 transcription factor on the YAP1 promoter. This prediction was fundamental to our study, as it guided the design of the three primers used to validate these binding sites in our ChIP-qPCR experiments (results shown in Figure 7E). In the revised version, we have significantly increased the figure's resolution and enlarged the font for the key predicted sequences to ensure they are clear and legible for proper assessment. Figure 8A is a protein-protein interaction network diagram sourced from the BioGRID database (Rose Oughtred et al., 2021), which summarizes proteins reported to interact with COL11A1. This analysis was crucial as it led us to identify the potential interaction between COL11A1 and SLC16A1 (MCT1), a key hypothesis in our work. To address your concern, we have replaced the previous figure with a high-resolution version.

Furthermore, to better highlight our findings, we have emphasized the interaction between COL11A1 and SLC16A1 with a thick red line, making it easier to identify. We believe these revisions have addressed your concerns, and the improved figures now provide the necessary clarity for a comprehensive evaluation. Thank you again for your guidance.

Revised figures:

Figure 7D

Figure 8A

References:

(1) Rauluseviciute, I., Riudavets-Puig, R., Blanc-Mathieu, R., Castro-Mondragon, J. A., Ferenc, K., Kumar, V., Lemma, R. B., Lucas, J., Chèneby, J., Baranasic, D., Khan, A., Fornes, O., Gundersen, S., Johansen, M., Hovig, E., Lenhard, B., Sandelin, A., Wasserman, W. W., Parcy, F., & Mathelier, A. (2024). JASPAR 2024: 20th anniversary of the open-access database of transcription factor binding profiles. *Nucleic acids research*, *52*(D1), D174–D182.

(2) Oughtred, R., Rust, J., Chang, C., Breitkreutz, B. J., Stark, C., Willems, A., Boucher, L., Leung, G., Kolas, N., Zhang, F., Dolma, S., Coulombe-Huntington, J., Chatr-Aryamontri, A., Dolinski, K., & Tyers, M. (2021). The BioGRID database: A comprehensive biomedical resource of curated protein, genetic, and chemical interactions. *Protein science : a publication of the Protein Society*, *30*(1), 187–200.

Comment #9: In Fig. 8E, the enlarged section: The notes of amino acid interactions are blurry. A similar illustration to Fig. 8F should be considered.

Reply:

Thanks for your suggestions. We have addressed this concern by completely remodeling the figure. The new Figure 8E and 8F have a significantly higher resolution, and we have created a new inset that clearly visualizes the key amino acid residues involved in the interaction. We are grateful for your constructive criticism, which has significantly improved the quality and clarity of our revised manuscript. Thank you again for your guidance.

Revised figures:

Figure 8E

Predicated docking model in silico

Figure 8F

Comment #10: The splinted excisional wound model in mice used in the current study is a widely recognized tool for studying wound healing, as it effectively simulates crucial processes like re-epithelialization and granulation tissue formation that occur in humans. However, it falls short of being a standard model for HS. A compelling 2022 systematic review of HS animal models (PMID: 35743999) concluded that no murine model, including those utilizing splinted wounds, can truly replicate the intricacies of human hypertrophic scars. This limitation arises from fundamental differences in skin anatomy and healing mechanisms between species. It is essential to acknowledge these differences and consider using more suitable models for the current research.

Reply: Thanks for your suggestions. We agree completely with the reviewer's assessment of the splinted excisional wound model in mice and the inherent limitations of using murine models for HS research. We have carefully read the systematic review by Rössler et al. (Stefan Rössler et al., 2022) that the reviewer kindly pointed out, and other meaningful reviews (Jialun Li et al., 2020; Mony, M. P. et al., 2023). We find its conclusions to be highly compelling and agree with its central thesis: currently, no single animal model perfectly replicates all the complex features of human HS, and therefore, **a universal "gold standard" model does not exist**. The choice of model must be carefully weighed against the specific research question being addressed.

Our study was designed to investigate the *in vivo* mechanistic role of a specific gene, *Mct1*, within the fibroblast population during the process of excessive scar formation. To rigorously test our hypothesis, a genetically knockout model that allows for cell-type-specific, inducible gene deletion was essential. As detailed in our manuscript, we utilized a *Mct1*^{fl/fl}*Col1a2*-

cre+ mouse model to achieve fibroblast-specific knockout of *Mct1*. **Moreover, the mouse remains the most feasible, established, and accessible mammalian system for creating such sophisticated conditional knockout models.** While larger animal models, such as the rabbit ear or porcine models, offer superior anatomical and physiological similarity to human skin (Stefan Rössler et al., 2022), the generation of conditional gene-edited lines in these species is associated with prohibitive costs, extended timelines, and significant technical challenges. **To maintain consistency and rigorously test our genetic hypothesis, the mouse model was the most appropriate choice.** Furthermore, while the human skin xenograft model on nude mice is a valuable tool (Stefan Rössler et al., 2022), it is unsuitable for our specific research question, as it does not permit the conditional knockout of a gene within the grafted human fibroblasts *in vivo*. Therefore, we selected the splinted full-thickness excisional wound model as the best available compromise. This model, by mechanically splinting the wound, prevents the contraction-based healing characteristic of rodents and instead promotes healing through granulation and re-epithelialization, which develops certain key features of human-like excessive scarring (e.g., increased collagen deposition, hypercellularity, and some dermal thickening). As our results show, this model was sufficiently sensitive to demonstrate both the effects of the *Mct1* knockout and the exacerbating phenotype caused by lactate administration, thus allowing us to validate our *in vitro* mechanistic findings.

In light of the reviewer's valid point, we have added this detailed explanation to the *Limitations of the current study* section in our revised manuscript to explicitly acknowledge these limitations and provide a clear justification for our model selection. Thank you again for your guidance.

Revised manuscript:

"Third, while our murine full-thickness excisional wound model partially recapitulates late-phase hypertrophic scar formation, interspecies differences in healing mechanisms persist: murine dermis containing the panniculus carnosus facilitates contractile healing through regenerative processes. However, the primary *in vivo* objective of our study was to elucidate the specific genetic function of fibroblast-expressed *Mct1* in a fibrotic healing context. This required a genetically knockout model allowing for conditional gene deletion, for which the mouse is the most established and feasible system. We chose the splinted wound model as the best available tool to investigate this specific molecular mechanism, while recognizing that future validation in larger animal models that more closely mimic human skin would be a valuable next step."

References:

- (1) Rössler, S., Nischwitz, S. P., Luze, H., Holzer-Geissler, J. C. J., Zrim, R., & Kamolz, L. P. (2022). In Vivo Models for Hypertrophic Scars-A Systematic Review. *Medicina (Kaunas, Lithuania)*, 58(6), 736.
 - (2) Li, J., Wang, J., Wang, Z., Xia, Y., Zhou, M., Zhong, A., & Sun, J. (2020). Experimental models for cutaneous hypertrophic scar research. *Wound repair and regeneration : official publication of the Wound Healing Society [and] the European Tissue Repair Society*, 28(1), 126–144.
 - (3) Mony, M. P., Harmon, K. A., Hess, R., Dorafshar, A. H., & Shafikhani, S. H. (2023). An Updated Review of Hypertrophic Scarring. *Cells*, 12(5), 678.
-

Comment #11: Delayed wound closure is not a defining characteristic of HS. Instead, its occurrence is usually influenced by factors such as infection, inadequate blood supply, or underlying conditions like diabetes. While delayed healing can be seen as a general indicator of wound healing and may lead to chronic wounds (e.g., ulcers), it does not specifically lead to HS. However, it can create conditions that predispose wounds to HS, especially in situations involving prolonged inflammation or mechanical tension. In contrast, the formation of HS is more directly related to the quality of healing, particularly excessive deposition of ECM, rather than the speed of wound closure. For instance, HS can develop in wounds that heal quickly if they are accompanied by exaggerated inflammatory and fibrotic responses. Therefore, it is important to carefully reconsider the interpretation of animal study data in light of these insights.

Reply:

Thanks for your suggestions. We completely agree with your expert assessment that HS is fundamentally a disorder of healing quality, characterized by excessive ECM deposition, rather than healing speed. The reviewer's point that delayed wound closure does not directly cause HS, but can create a predisposing pro-fibrotic environment, is well-taken and has allowed us to refine the interpretation of our findings. Furthermore, this comment allows us to better highlight a central novel aspect of our study. As the reviewer astutely noted, a prolonged inflammatory phase is a key factor that predisposes wounds to HS. **The significance of our research lies in expanding the understanding of the macrophage's role in this process.** Traditionally, macrophages are viewed as drivers of fibrosis primarily through the secretion of classical cytokines and growth factors (Geoffrey C Gurtner et al.,

2008; Marc G Jeschke et al., 2023). Our work proposes a complementary and critical axis: macrophages also act as key metabolic regulators of the wound microenvironment through the secretion of lactate. This perspective shifts the view of lactate from a mere metabolic byproduct to a pivotal signaling molecule that directly orchestrates fibroblast phenotype and subsequent fibrosis. **Therefore, we interpret the delayed wound closure observed in our model not as a direct cause of HS, but as a clinical manifestation of this persistent, lactate-rich, pro-inflammatory, and pro-fibrotic microenvironment orchestrated by macrophages.** When we inhibit this axis (via genetic ablation of *Mct1* or pharmacological inhibition with AZD3965), we are targeting this fundamental metabolic crosstalk. This leads to two parallel improvements: (1) Improved healing speed: A more rapid wound closure, which we interpret as a sign of a more efficiently resolved, less pathologically active microenvironment. (2) Improved healing quality: A reduction in fibrotic gene expression (*Col1a1*, *Col3a1*), decreased collagen deposition, and attenuated scar formation, by directly cutting off the lactate signal to fibroblasts.

We acknowledge that our original manuscript may have conflated these two outcomes. Based on the reviewer's valuable guidance, we have carefully revised the manuscript, particularly in the Results and Discussion sections, to present this more precise interpretation. We now clarify that accelerated wound closure and reduced scarring are both consequences of targeting the fundamental lactate-driven fibrotic mechanism, rather than suggesting that one causes the other. We thank the reviewer again for helping us to significantly improve the clarity and impact of our manuscript.

Revised manuscript:

“Collectively, these results suggested that the pro-fibrotic signaling driven by lactate and MCT1 creates a pathological microenvironment that simultaneously impairs the rate of wound closure and drives the excessive fibrotic gene expression characteristic of hypertrophic scarring, mediated by increased H3K231a levels in fibroblasts.”

“Critically, targeting this pathway via fibroblast-specific *Mct1* deletion or pharmacological inhibition (AZD3965) mitigated the core pathology. This intervention resulted in two concomitant benefits: an acceleration of wound closure, likely reflecting a more rapid resolution of the pro-fibrotic inflammatory phase and a reduction in the excessive collagen deposition that defines the quality of the scar.”

References:

(1) Gurtner, G. C., Werner, S., Barrandon, Y., & Longaker, M. T. (2008). Wound repair and regeneration. *Nature*, 453(7193), 314-321.

(2) Jeschke, M. G., Wood, F. M., Middelkoop, E., Bayat, A., Teot, L., Ogawa, R., et al. (2023). Scars. *Nature reviews Disease primers*, 9(1), 64.

Comment #12: Lines 400-402, 'According to the specific structure of the skin shown by HE staining, collagen arrangement was smoother and more orderly ... '

Firstly, the description of 'the specific structure' requires more detail to enhance clarity and understanding.

Secondly, incorporating established methods to quantitatively assess the smoothness and orderliness of collagen fibers would strengthen the supporting evidence for the statement.

Reply:

Thanks for your suggestions. We are sorry for the inappropriate description of the histological findings and the absence of quantitative evidence for collagen arrangement. We have carefully revised the manuscript according to your suggestions. (1) Regarding the description of H&E staining: We acknowledge that the phrase 'the specific structure' was ambiguous. To address this, we have revised the text to provide a more detailed and professional description of the histopathological features observed in the H&E-stained sections. We now describe specific changes in the epidermis and dermis, including epidermal thickness, the presence or absence of rete ridges, inflammatory cell infiltration, hair follicle, and the overall dermal architecture in different experimental groups. We believe these changes have greatly enhanced the clarity of our findings. (2) Regarding the quantitative assessment of collagen fibers: We concur that our qualitative description of collagen arrangement as 'smoother and more orderly' should be supported by objective, quantitative data. To achieve this, we have performed a quantitative analysis of collagen fiber alignment on our Masson's trichrome-stained histological images using the widely accepted image analysis software ImageJ with the Directionality plugin (Alberto Sensini et al., 2018). This method allowed us to calculate the dispersion index of collagen fibers, where a lower value indicates a more orderly and aligned arrangement. The new quantitative data demonstrates a significantly lower dispersion of collagen fibers in the *Mct1* knockout group compared to the control group, which corroborates our initial qualitative observation. These results, along with the corresponding statistical analysis, have been added to the manuscript as a new figure panel (Figure S11A and S11C) and are

described in the Results section. We are confident that these revisions have substantially improved the rigor and quality of our manuscript, and we are grateful for the opportunity to strengthen our work based on your expert guidance. Thank you again for your guidance.

Revised manuscript:

“Histological analysis with H&E staining revealed that wounds in the *Mct1* knockout group exhibited a more organized skin architecture, characterized by a well-formed epidermis with rete ridges and signs of hair follicle regeneration, and a dermis with reduced inflammatory cell infiltration compared to the control group. In contrast, the control and NaLac-treated groups showed features of hypertrophic scarring, including a thickened, flattened epidermis and a disorganized dermal structure (Figure 9D). Masson staining showed obviously reduced blue collagen deposition and collagen fibers were arranged in a smoother, more orderly pattern in the *Mct1^{fl/fl}Colla2-cre⁺* compared to *Mct1^{fl/fl}Colla2-cre⁻* mice (Figure 9D). Quantitative analysis confirmed these observations, showing a significantly reduced collagen area and dispersion index—indicating more orderly collagen alignment—in the *Mct1* knockout group (Figure 9E and S10A).”

Revised figures:

Figure S8A

Figure S8C

References:

(1) Sensini, A., Gualandi, C., Zucchelli, A., Boyle, L. A., Kao, A. P., Reilly, G. C., Tozzi, G., Cristofolini, L., & Focarete, M. L. (2018). Tendon Fascicle-Inspired Nanofibrous Scaffold of Polylactic acid/Collagen with Enhanced 3D-Structure and Biomechanical Properties. *Scientific reports*, 8(1), 17167.

Comment #13: The ratio of collagen I to collagen III undergoes a dynamic shift during the wound healing process, reflecting the stages of tissue repair and remodeling. Thus, the noted reduction in the collagen I/III ratio in Figs. 9 and 10 may be just a secondary effect of delayed healing, which should not be overrated.

Reply:

Thanks for your suggestions. We agree entirely that the collagen I/III ratio is a dynamic marker that changes significantly throughout the phases of wound healing. Your point that a change in this ratio could reflect the healing stage rather than a direct therapeutic effect is well-taken, and we appreciate the opportunity to clarify our interpretation.

Our analysis was conducted at day 21 post-wounding, a time point that is well within the remodeling phase of healing, where collagen III, prevalent in the early proliferative phase, is typically replaced by the stronger, more mature collagen I (Oscar A Peña et al., 2024; Pauline D H M Verhaegen et al., 2009). While a shift towards collagen I is part of normal remodeling,

an excessively high collagen I/III ratio, coupled with disorganized collagen fiber arrangement, is a well-established hallmark of pathological fibrosis and hypertrophic scar (Ying Zhang et al., 2023; Qing Zhang et al., 2022).

Therefore, our finding that *Mct1* ablation (Figure 9) or pharmacological inhibition with AZD3965 (Figure 10) leads to a reduced collagen I/III ratio at day 21 should be interpreted not as a sign of delayed healing, **but as a shift away from a pro-fibrotic, pathological trajectory towards a more regenerative healing process that better resembles the composition of normal dermis.** This interpretation is strongly supported by our complementary findings presented in Figures 9 and 10: (1) Accelerated wound closure: Both genetic ablation and pharmacological inhibition of MCT1 led to significantly faster wound closure (Figure 9C and 10B), which contradicts the idea of delayed healing. The lactate-treated groups, which exhibited a higher collagen I/III ratio, showed delayed wound closure. (2) Reduced overall collagen deposition: Masson and Sirius red staining, and the subsequent quantification (Figure 9F and 10E), clearly show a significant reduction in total collagen content in the MCT1-inhibited groups. (3) Improved collagen organization: As observed in the Masson staining images and quantitative analysis of dispersion index, the collagen fibers in the treated groups were more organized, resembling normal skin architecture, whereas the control/lactate-treated groups showed thick, disorganized collagen bundles characteristic of scar tissue (Figure 9D, 10C, S11A and S11C). (4) Reduced myofibroblast presence: We also observed a decrease in α -SMA expression (Figure 9H, 9I and 10G) in the *Mct1* knockout and inhibition groups, indicating a reduction in the key cell type responsible for excessive matrix deposition and scar contraction. In summary, the reduced collagen I/III ratio, when viewed in the context of accelerated healing, reduced total collagen, improved collagen organization, and decreased myofibroblast markers, provides compelling evidence for an anti-fibrotic effect. Our intervention does not delay healing but rather modulates it to produce a higher-quality, less-scarred tissue.

To clarify this important point for the reader, we have revised the manuscript to discuss these results with greater nuance, explicitly contextualizing the collagen I/III ratio as a marker of healing quality, not just healing stage. Thank you again for your constructive feedback.

Revised manuscript:

“The collagen I to III ratios of groups were reduced in *Mct1* knockdown mice compared to the control mice (Figure 9D and 9G). Meanwhile, NaLac treatment resulted in enhanced collagen synthesis, disorganized collagen alignment, and elevated collagen I to III ratios, as

notably observed in the histological analysis (Figure 9D-G). These data indicated a shift from a fibrotic to a more regenerative healing phenotype in *Mct1* knockdown mice.”

References:

(1) Peña, O. A., & Martin, P. (2024). Cellular and molecular mechanisms of skin wound healing. *Nature reviews. Molecular cell biology*, 25(8), 599–616.

(2) Verhaegen, P. D., van Zuijlen, P. P., Pennings, N. M., van Marle, J., Niessen, F. B., van der Horst, C. M., & Middelkoop, E. (2009). Differences in collagen architecture between keloid, hypertrophic scar, normotrophic scar, and normal skin: An objective histopathological analysis. *Wound repair and regeneration : official publication of the Wound Healing Society [and] the European Tissue Repair Society*, 17(5), 649–656.

(3) Zhang, Y., Wang, S., Yang, Y., Zhao, S., You, J., Wang, J., Cai, J., Wang, H., Wang, J., Zhang, W., Yu, J., Han, C., Zhang, Y., & Gu, Z. (2023). Scarless wound healing programmed by core-shell microneedles. *Nature communications*, 14(1), 3431.

(4) Zhang, Q., Shi, L., He, H., Liu, X., Huang, Y., Xu, D., Yao, M., Zhang, N., Guo, Y., Lu, Y., Li, H., Zhou, J., Tan, J., Xing, M., & Luo, G. (2022). Down-Regulating Scar Formation by Microneedles Directly via a Mechanical Communication Pathway. *ACS nano*, 16(7), 10163–10178.

Comment #14: Line 475: The reference to Zhang et al. should be provided.

Reply: Thanks for your suggestions. We would like to express our gratitude to the reviewer for pointing out this omission. We have now added the requested citation for Zhang et al. in the manuscript.

Reviewer #2:

In the manuscript titled “Lactate derived from macrophages drives skin dermal fibroblasts phenotypic remodeling via MCT1-primed histone H3 lysine 23 lactylation in hypertrophic scar,” the authors investigate how matrix stiffness–activated macrophages produce excess lactate, which is shuttled into dermal fibroblasts via MCT1. This influx promotes site-specific histone H3K23 lactylation (H3K23la), leading to upregulation of profibrotic genes HEY2 and COL11A1. Through CUT&Tag, RNA-seq, molecular simulations, and both genetic (fibroblast-specific Mct1 knockout) and pharmacological (AZD3965) interventions in murine wound models, they demonstrate that disrupting the MCT1–H3K23la–HEY2/COL11A1 feedback loop attenuates hypertrophic scar formation. The presented findings are new and supported by the experimental data. The reviewer has several comments listed below.

Comment #1: The immunofluorescence images in Figure 1F show MCT1 colocalization with α -SMA–positive myofibroblasts, but no quantitative colocalization analysis (e.g., Pearson’s coefficient) is provided. Please include such metrics to substantiate the claim.

Reply:

Thanks for your suggestions. We agree that a quantitative analysis of the colocalization between MCT1 and α -SMA is essential to substantiate our claim and significantly strengthen our findings. Following the reviewer's recommendation, we have performed a quantitative colocalization analysis on the immunofluorescence images from Figure 1F using Pearson's correlation coefficient analysis in software ImageJ. The results confirm a significantly higher degree of colocalization between MCT1 and α -SMA in HS tissues compared to NS tissues. We have now included these quantitative results as a new panel of Figure 1H. We have also updated the manuscript text in the Results section and the legend for Figure 1 to refer to this new analysis. We believe these additions have greatly improved the manuscript by providing robust, quantitative evidence for our observations. We are grateful for the opportunity to enhance the quality of our work. Thank you again for your guidance.

Revised figures:

Figure 1H

Revised manuscript:

“To quantitatively validate this observation, we calculated the Pearson's correlation and Overlap coefficient, which revealed a significantly higher level of colocalization between MCT1 and α -SMA in HS tissues compared to NS tissues (Figure 1F).”

Comment #2: The rationale for using 10 ng/mL TGF- β is not explained. Cite relevant literature or provide preliminary data to justify this concentration.

Reply:

Thank you for your valuable feedback and for highlighting the need to clarify our choice of TGF- β concentration. We agree that the rationale for using 10 ng/mL of TGF- β 1 was not sufficiently explained in the original manuscript, and we appreciate the opportunity to address this. This concentration has been shown to be optimal for stimulating fibroblast proliferation, differentiation into myofibroblasts, and excessive ECM deposition, which are key events in the pathogenesis of hypertrophic scars. For instance, several studies investigating the mechanisms of hypertrophic scar have successfully used 10 ng/mL of TGF- β to activate human hypertrophic scar fibroblasts (HSFs) and normal skin fibroblasts (NSFs) (Keun Jae Ahn et al., 2022; Amy S Colwell et al., 2005). Furthermore, this concentration is commonly employed in broader fibrosis research to create reliable *in vitro* models of fibrosis suitable for mechanistic studies and screening of anti-fibrotic agents (Ruijuan Guan et al., 2022; Li Xiao

et al., 2012; Qihe Xu et al., 2007). Based on this strong precedent in the field, we selected this concentration to ensure a potent and reproducible fibrotic stimulation in our experimental system, particularly for activating macrophages (THP-1 cells) to study their subsequent effects on fibroblasts, as described in our manuscript.

To address your concern, we have revised the Methods section of our manuscript to include this justification and have cited relevant literature. We believe this addition strengthens the methodological foundation of our study. Thank you again for your constructive suggestion.

Revised manuscript:

“This concentration was selected as a well-established and widely used concentration in the literature for effectively inducing a fibrotic response in in vitro models of hypertrophic scarring and fibrosis^{62,63}.”

References:

- (1) Ahn, K. J., & Kim, J. S. (2022). TGF- β 1 upregulates Sar1a expression and induces procollagen-I secretion in hypertrophic scarring fibroblasts. *Open medicine (Warsaw, Poland)*, 17(1), 1473–1482.
- (2) Colwell, A. S., Phan, T. T., Kong, W., Longaker, M. T., & Lorenz, P. H. (2005). Hypertrophic scar fibroblasts have increased connective tissue growth factor expression after transforming growth factor-beta stimulation. *Plastic and reconstructive surgery*, 116(5), 1387–1392.
- (3) Guan, R., Yuan, L., Li, J., Wang, J., Li, Z., Cai, Z., Guo, H., Fang, Y., Lin, R., Liu, W., Wang, L., Zheng, Q., Xu, J., Zhou, Y., Qian, J., Ding, M., Luo, J., Li, Y., Yang, K., Sun, D., ... Lu, W. (2022). Bone morphogenetic protein 4 inhibits pulmonary fibrosis by modulating cellular senescence and mitophagy in lung fibroblasts. *The European respiratory journal*, 60(6), 2102307.

(4) Xiao, L., Du, Y., Shen, Y., He, Y., Zhao, H., & Li, Z. (2012). TGF-beta 1 induced fibroblast proliferation is mediated by the FGF-2/ERK pathway. *Frontiers in bioscience (Landmark edition)*, 17(7), 2667–2674.

(5) Xu, Q., Norman, J. T., Shrivastav, S., Lucio-Cazana, J., & Kopp, J. B. (2007). In vitro models of TGF-beta-induced fibrosis suitable for high-throughput screening of antifibrotic agents. *American journal of physiology. Renal physiology*, 293(2), F631–F640.

Comment #3: The manuscript proposes that lactate enters fibroblasts via MCT1 to induce H3K231a and activate HEY2 and COL11A1 transcription, but lacks details on the dynamic regulation of this process. For example, the dose–response relationship between lactate concentration and H3K231a levels has not been fully characterized.

Reply:

Thanks for your suggestions. We agree that characterizing the dynamic relationship between lactate concentration and H3K231a is crucial for strengthening the core argument of our study. To address your concern, we have performed additional experiments. Specifically, we treated normal skin fibroblasts (NSFs) with a gradient of sodium lactate concentrations (0, 5, 10, 15, 20 and 25 mM) and assessed the levels of H3K231a by Western blot. Our new data reveal a clear dose-dependent increase in H3K231a level, with the induction of H3K231a becoming significant at 5 mM and peaking at approximately 10-15 mM. These findings directly demonstrate that extracellular lactate levels dynamically regulate histone lactylation in fibroblasts. We have incorporated these new findings into the manuscript. The text in the Results section has been revised, and we have added a new supplementary figure (Figure S5A) to present this dose-response data. We believe these additions substantially strengthen our manuscript by providing the quantitative evidence you rightly pointed out was missing. Thank you again for your constructive suggestion.

Revised figures:

Figure S4A

Revised manuscript:

“Furthermore, we also found that there was a clear dose-dependent increase in H3K231a level under the different concentration of NaLac (Figure S5A).”

Comment #4: The authors state that 2 kPa and 50 kPa hydrogels were used to model physiological and pathological matrix stiffness, but no supporting references or experimental stiffness measurements are provided. Please cite precedent studies or supply experiment data.

Reply:

Thanks for your suggestions. We agree that providing justification for the selected hydrogel stiffness values is crucial for the manuscript. We apologize for this oversight and have revised the manuscript to include appropriate citations that support our choice of 2 kPa and 50 kPa to model physiological and pathological matrix stiffness, respectively. The selection of these values is based on well-established precedents in the literature that characterize the mechanical properties of normal and fibrotic skin (Jiahao He et al., 2023). Normal, healthy dermal tissue typically exhibits a Young's modulus in the low kPa range. Our use of 2 kPa hydrogels to mimic a "physiologically compliant" substrate is consistent with findings from multiple studies investigating dermal fibroblast mechanobiology. In contrast, fibrotic tissues, such as hypertrophic scars and keloids, are characterized by excessive extracellular matrix deposition, leading to a significant increase in tissue stiffness. The use of hydrogels

with a Young's modulus of 50 kPa to simulate a "pathologically stiff" or fibrotic microenvironment is a well-established *in vitro* model. For instance, He et al. in their study on mechanical stiffness and skin fibrosis, utilized the exact same values of 2 kPa and 50 kPa to represent normal and fibrotic skin, respectively (Jiahao He et al., 2023). Furthermore, Lin et al. investigated the role of matrix stiffness in benign prostatic hyperplasia and fibrosis, similarly using soft substrates in the low kPa range (~1-2 kPa) and rigid substrates in the tens of kPa range (~50 kPa) to model normal versus pathological conditions, reinforcing the principle of our experimental design (Dongxu Lin et al., 2024). We have now incorporated these and other relevant citations into the manuscript to provide clear support for our experimental model. We believe these additions address your concern and strengthen the methodological foundation of our study. Thank you again for your valuable feedback.

References:

- (1) He, J., Fang, B., Shan, S., & Li, Q. (2023). Mechanical stiffness promotes skin fibrosis through Piezo1-mediated arginine and proline metabolism. *Cell death discovery*, 9(1), 354.
- (2) Lin, D., Luo, C., Wei, P., Zhang, A., Zhang, M., Wu, X., Deng, B., Li, Z., Cui, K., & Chen, Z. (2024). YAP1 Recognizes Inflammatory and Mechanical Cues to Exacerbate Benign Prostatic Hyperplasia via Promoting Cell Survival and Fibrosis. *Advanced science (Weinheim, Baden-Wurtemberg, Germany)*, 11(5), e2304274.

Reviewer #3

In their manuscript, entitled “Lactate derived from macrophages drives skin dermal fibroblasts’ phenotypic remodeling via MCT1-primed histone H3 lysine 23 lactylation in hypertrophic scar,” Yuan et al. investigate increased lactate production in the hypertrophic skin and its influence on fibroblast function in fibrosis and wound healing. The authors further determined that MCT1 functions to transport lactate into fibroblasts, where it is utilized for histone lactylation. This, in turn, activates the transcription of profibrotic genes HEY2 and COL11A1, which activate YAP1 expression and aggravate lactate accumulation, respectively. Finally, the authors test the relevance of MCT1 expression in skin fibroblasts using a wound healing model, employing genetic depletion and compound blocking.

This reviewer is deeply concerned about the novelty of the findings, the technical quality of the presented data, and the technical quality of the manuscript. The role of MCT1 in fibrosis and keloid fibroblasts has been known for some time (Jinhua Feng et al., 2024; David R. Ziehr et al., 2024 preprint; Kyounghee Min, 2023; Jing-Jing Gu 2025, and many others), as well as the role of lactylation in fibrosis (Huanwei Liang et al., 2025 review). Therefore, the novelty of the study results is limited, if any.

The presented manuscript is generally poorly written, contains numerous grammatical errors, and references data that are not present in the figures. Additionally, it includes too many figures that do not contribute to the study's quality.

Reply:

Thank you for your time and for providing a detailed critique of our manuscript. We appreciate the opportunity to respond to the concerns raised and to improve our work. While we understand the reviewer's reservations, we believe there may be a misunderstanding of the specific context and the novel mechanistic details our study provides. We have addressed each of the concerns below.

Regarding the novelty of the findings

We thank the reviewer for highlighting the existing literature. We agree that metabolic reprogramming, including the roles of MCT1 and histone lactylation, are recognized as important factors in various fibrotic diseases. However, we respectfully assert that our study provides a significant and novel contribution to the field by elucidating a previously uncharacterized, specific, and complete mechanistic pathway in the distinct context of hypertrophic scar. Our novelty is multi-faceted and lies in the integration and specificity of our findings: (1) A novel and specific axis in hypertrophic scar: While the cited papers discuss MCT1 in broader fibrosis or keloids, our study is the first to establish the complete axis in HS: **Stiff microenvironment → Macrophage glycolysis → Lactate secretion → Fibroblast MCT1 uptake → Histone H3K23 lactylation → HEY2/COL11A1 upregulation → YAP1/SMAD2 activation & Positive feedback.** To our knowledge, this integrated pathway has never been described before. The papers cited by the reviewer discuss parts of this puzzle in different contexts (e.g., MCT1 in keloid metabolism or general lactylation in fibrosis), but none have connected these specific molecular players in this precise sequence to drive HS pathogenesis. (2) Identification of a specific histone mark (H3K23la): The reviewer mentions a review on "lactylation in fibrosis." Our work moves beyond this by providing new, primary data identifying H3K23la as the specific and key epigenetic modification that is significantly upregulated in HS myofibroblasts. We demonstrate through CUT&Tag analysis that H3K23la enrichment is directly responsible for the transcriptional activation of the specific profibrotic genes *HEY2* and *COL11A1*. This level of specificity—linking a single histone mark to specific gene promoters in HS—is a core novel finding of our study. (3) Elucidation of the upstream cellular source and trigger: A major novel contribution of our work is the identification of **macrophages, activated by a stiff mechanical microenvironment, as the primary source of lactate** that fuels fibroblast remodeling in HS. Previous studies have often focused on fibroblast-intrinsic metabolic changes (Xinxian Meng et al., 2022). Our findings reveal a critical intercellular crosstalk (macrophage-to-fibroblast) that initiates the fibrotic cascade, redefining the upstream triggers in HS pathology. (4) Discovery of a self-perpetuating positive feedback loop: Our study uncovers a novel positive feedback mechanism where a downstream target, **COL11A1, physically interacts with and stabilizes MCT1**, thereby enhancing further lactate transport and amplifying the fibrotic signal. This

self-reinforcing loop (Lactate → H3K231a → COL11A1 → MCT1 stabilization → more Lactate influx) provides a new paradigm for understanding the persistence and progression of hypertrophic scars.

To better emphasize these points, we have revised the Introduction and Discussion sections and cited relatively new literature about MCT1 and histone lactylation. We now more explicitly contextualize our work within the existing literature and then clearly articulate the specific, novel contributions that our detailed mechanistic investigation provides to the field of skin fibrosis and hypertrophic scar.

Regarding the technical quality of the manuscript and figures

We sincerely apologize that the manuscript's language did not meet the reviewer's standards. The manuscript was professionally edited by Language Editing Services prior to submission to ensure linguistic accuracy. However, we acknowledge that scientific clarity goes beyond grammar. We have meticulously re-read the entire manuscript to improve sentence structure, enhance clarity, and ensure the logical flow of our arguments is easy to follow. We have focused on simplifying complex sentences to make our findings more accessible.

Furthermore, we have performed a rigorous check of the entire manuscript, cross-referring every statement in the results section with its corresponding figure panel and legend. We have corrected any discrepancies to ensure that all claims are directly and clearly supported by the data presented. Finally, we appreciate the reviewer's feedback on the manuscript's structure. We believe that each figure is essential for building our comprehensive story, from the initial observation of metabolic dysregulation (Figure 1), to identifying the cellular source (Figure 2), to systematically dissecting the entire molecular pathway (Figures 3-8), and finally, to validating our findings using *in vivo* genetic and pharmacological models (Figures 9-10). The logical progression from *in vitro* mechanism to *in vivo* relevance necessitates this level of detail. However, taking the reviewer's comment into consideration, we have re-evaluated all supplementary figures and have consolidated some panels to improve the manuscript's focus without compromising the data's integrity.

In summary, we are confident that our study presents a significant and novel contribution by delineating a complete and specific metabolic-epigenetic axis that drives hypertrophic scar formation. We have taken the reviewer's constructive feedback seriously and have thoroughly revised the manuscript to improve its clarity, accuracy, and impact. We thank you again for your valuable time and consideration.

References:

(1) Meng, X., Yu, Z., Xu, W., Chai, J., Fang, S., Min, P., Chen, Y., Zhang, Y., & Zhang, Z. (2022). Control of fibrosis and hypertrophic scar formation via glycolysis regulation with IR780. *Burns & trauma*, 10, tkac015.

Comment #1: The majority of the paper utilizes an *in vitro* system to investigate a complex and multifaceted disease that affects complex tissue, while only selected experiments employ tissue samples. It is unclear why these shifts in experimental systems are made, as they are by no means beneficial to the study.

Reply:

We sincerely thank the reviewer for this insightful comment and appreciate the opportunity to clarify our experimental rationale. We agree that hypertrophic scar is a complex, multifactorial process involving intricate crosstalk between various cell types within a dynamic extracellular matrix. It was precisely this complexity that guided our decision to employ a multi-tiered experimental approach, progressing logically from controlled *in vitro* systems to clinically relevant human tissues and finally to functional *in vivo* models. We believe this strategic progression is a key strength of our study, allowing us to build a robust, evidence-based narrative from fundamental mechanism to potential therapeutic application.

We would like to elaborate on the specific purpose and necessity of each experimental system: (1) **The necessity of the *in vitro* system for mechanistic dissection:** Our primary goal was to dissect the specific molecular and cellular drivers of fibroblast activation in HS.

The complex microenvironment of scar tissue, with its multitude of cell types and signaling molecules, makes it exceedingly difficult to isolate and study the contribution of a single factor, such as mechanical stiffness, on a specific cell type. Our *in vitro* system, utilizing PDMS hydrogels of varying stiffness (2 kPa vs. 50 kPa), was essential to answer a critical initial question: **What is the primary cellular source of the elevated lactate we observe in HS tissue?** By culturing the three main dermal cell populations (macrophages, fibroblasts, and endothelial cells) separately on these defined substrates, we were able to unequivocally demonstrate that macrophages, when subjected to a pathologically stiff environment, are the predominant source of lactate (Figure 2). Furthermore, the *in vitro* system was indispensable for testing direct causal relationships. For instance, using conditioned medium (CM) from macrophages cultured on stiff substrates, we could directly test its effect on fibroblasts. The subsequent use of the MCT1 inhibitor (AZD3965) and the glycolysis inhibitor (Oxamate) allowed us to prove that macrophage-derived lactate, transported via MCT1, was the key mediator of fibroblast activation and phenotypic remodeling (Figure 3). Similarly, our siRNA-mediated knockdown experiments (e.g., silencing of *MCT1*, *HEY2*, and *COL11A1*) were critical for mapping the downstream epigenetic (H3K23la) and transcriptional pathways, which can only be performed in a controlled cell culture environment. (2) **The role of human tissue samples for clinical validation:** While our *in vitro* work provided strong mechanistic evidence, its relevance hinges on whether these pathways are active in human disease. Therefore, we used human hypertrophic scar and normal skin tissues to validate our findings and bridge the translational gap. Our analyses of patient-derived tissues confirmed that the key components of our proposed axis are indeed dysregulated in HS. We demonstrated elevated levels of lactate, increased expression of MCT1, and a specific and significant upregulation of histone H3K23 lactylation in HS tissue compared to normal skin (Figure 1, Figure 4). We further showed that the downstream targets, *HEY2* and *COL11A1*, were also significantly elevated in patient scar tissue (Figure 5). This step was vital to confirm that the mechanism we dissected *in vitro* is a genuine feature of the human pathology. **Crucially, we also utilized these patient samples to directly test our key *in vitro* conclusion regarding the cellular origin of lactate (Question 7).** First, through multicolor immunofluorescence staining of human HS and NS sections, we confirmed that glycolytic pathway activity was

significantly increased specifically within the macrophage population (Figure 2I and 2J). In contrast, endothelial cells showed no statistically significant changes, mirroring our *in vitro* observations (Figure S3D and S3E). Second, to further substantiate this, we isolated primary macrophages, endothelial cells, and fibroblasts directly from both normal skin and hypertrophic scar tissues. Direct measurement of extracellular and intracellular lactate levels in these primary cells revealed that macrophages isolated from HS exhibited the most substantial and significant increase in lactate content, and elevated ECAR level (Figure 2G and 2H). Furthermore, we have performed a new *in vivo* experiment using our full-thickness skin defect mouse model. In this experiment, we selectively depleted macrophages during the wound healing process to directly assess their contribution to lactate production. Specifically, we administered clodronate liposomes subcutaneously starting on day 7 post-injury to deplete macrophages, with a control group receiving saline injections. On day 21, wound tissues were harvested for analysis. Immunofluorescence staining for the macrophage marker F4/80 confirmed a significant reduction in macrophage numbers in the scar tissues of clodronate-treated mice compared to controls (Figure S2F). Crucially, lactate concentration was markedly decreased in the scar tissue of macrophage-depleted mice (Figure S2G). These experiments, performed on clinically relevant human tissues and animal model, provide compelling evidence that corroborates our *in vitro* findings. They strongly support our conclusion that macrophages are the principal source of the lactate-rich microenvironment that drives fibrotic remodeling in hypertrophic scars. (3) **The importance of the *in vivo* model for functional and therapeutic proof-of-concept:** Finally, to move beyond correlation and establish causality in a complex biological system, we utilized a murine full-thickness wound model. This *in vivo* model allowed us to test the functional necessity of our pathway in the context of tissue repair and scarring. By employing a fibroblast-specific knockout of *Mct1* (*Mct1^{fl/fl}Colla2-cre⁺* mice), we demonstrated that ablating lactate transport specifically in fibroblasts leads to accelerated wound healing and attenuated scar formation (Figure 9). This genetic evidence provides powerful, direct proof of our hypothesis. The *in vivo* model was also essential for testing the therapeutic viability of targeting this pathway. The pharmacological inhibition of MCT1 with AZD3965 in our wound model recapitulated the positive effects of the genetic knockout, significantly reducing scar features (Figure 10).

This provides a critical preclinical proof-of-concept that MCT1 is a druggable target for HS therapy.

In summary, our experimental design was intentionally structured to investigate this complex disease in a logical and stepwise manner. The *in vitro* system was necessary to dissect the fundamental cellular and molecular mechanisms with high precision. The human tissue samples were essential to validate the clinical relevance of these mechanisms. Lastly, the *in vivo* animal model was crucial to confirm the functional importance of the pathway and to provide a preclinical proof-of-concept for a novel therapeutic strategy. Therefore, we respectfully argue that these shifts in experimental systems are not only beneficial but are in fact essential to the study's design. This multi-pronged approach allows us to build a comprehensive, multi-level narrative from molecular mechanism to clinical observation and, ultimately, to therapeutic implication. We hope this explanation has adequately addressed the reviewer's concerns. We would be pleased to incorporate a more detailed explanation of our experimental strategy into the revised manuscript if deemed appropriate. Thank you again for your valuable guidance.

Revised manuscript:

“Our study establishes a novel axis linking metabolic reprogramming and epigenetic regulation in hypertrophic scar pathogenesis. First and foremost, we redefine the upstream trigger of fibroblast activation by identifying mechanically stressed macrophages as the key lactate producers. The stiff microenvironment induces macrophages to adopt a hyperglycolytic phenotype, producing lactate that is shuttled into fibroblasts via MCT1. Second, this lactate influx primes H3K231a, activating profibrotic genes *HEY2* and *COL11A1*. *HEY2* amplifies fibrotic signaling via YAP1/SMAD2, while *COL11A1* stabilizes MCT1 to enhance lactate transport, creating a self-reinforcing loop that perpetuates fibrosis. Furthermore, the process initiates in the stiff HS microenvironment, which induces macrophages to produce excessive lactate. Critically, targeting this pathway via fibroblast-specific *Mct1* deletion or pharmacological inhibition (AZD3965) mitigated the core

pathology. This intervention resulted in two concomitant benefits: an acceleration of wound closure, likely reflecting a more rapid resolution of the pro-fibrotic inflammatory phase and a reduction in the excessive collagen deposition that defines the quality of the scar. This work provides the first evidence that macrophage-derived lactate fuels HS progression through MCT1-dependent histone lactylation, redefining lactate as a metabolic-epigenetic mediator rather than a passive byproduct. These findings position MCT1 as a therapeutic target and establish histone lactylation as a novel epigenetic driver of HS progression.”

Comment #2: In the first figure, authors proceed directly to measuring lactate levels in the hypertrophic skin, providing little justification for why they chose to present precisely this metabolite and not assess the overall metabolomic profile of hypertrophic versus normal skin. The authors should provide an extended metabolomic profile of hypertrophic versus normal skin, allowing for a comparison of the presence of different metabolites in the skin.

Reply:

We thank you for this insightful comment and for the opportunity to clarify our scientific rationale. We agree completely that providing a broader metabolomic context is crucial for justifying our focus on lactate and significantly strengthens the introduction to our findings.

In response to your suggestion, we have performed and now included data from a non-targeted metabolomic analysis of hypertrophic scar (HS) and normal skin (NS) tissues. This initial exploratory analysis revealed a significant and distinct metabolic reprogramming in HS tissues. Specifically, we found that key intermediates of the TCA cycle, such as citric acid, were significantly decreased in HS. Conversely, metabolites of the glycolytic pathway, such as phosphoenolpyruvic acid, were significantly increased (Figure S1A-S1C). This compelling evidence strongly indicates a metabolic shift in hypertrophic scar away from oxidative phosphorylation and towards aerobic glycolysis. As lactate is the primary end-product of aerobic glycolysis, these metabolomic data provided a robust, data-driven rationale for our subsequent, more targeted investigation into lactate accumulation and its functional role in HS pathogenesis, as presented in Figure 1.

To incorporate this important context into the manuscript, we have made the following changes. We have created a new Supplementary Figure (Figure S1) to present the key findings from our non-targeted metabolomic analysis (including a volcano plot of differential metabolites and levels of citrate and phosphoenolpyruvate). We have substantially revised the first section of the Results to begin with these metabolomic findings, establishing the metabolic shift to glycolysis as the logical basis for our subsequent focus on lactate. We have updated the Discussion to integrate these findings, further highlighting the importance of metabolic reprogramming in HS and strengthening our comparison with the metabolism of keloids. We have also made our raw metabolomics data publicly available in the GSA-Human database as referenced in the manuscript, ensuring full transparency. We believe these additions directly address your concern and have made the manuscript's narrative clearer and more scientifically rigorous. We are grateful for your valuable feedback, which has undoubtedly improved the quality of our paper. Thank you again for your valuable guidance.

Revised figures:

Figure S1

Revised manuscript:

“To characterize the metabolic phenotype, non-targeted metabolomic analysis of HS and healthy tissues revealed a distinct metabolic profile in patients (Figure S1A; Supplemental data 1). HS tissue exhibited a significant decrease in the key TCA cycle intermediate, citric acid, and a concomitant increase in the glycolytic metabolite, phosphoenolpyruvic acid (Figure S1A-S1C).”

“This concurrent upregulation of the glycolytic pathway and downregulation of key OXPHOS components provides strong evidence of a metabolic switch towards aerobic glycolysis in the HS microenvironment.”

“We have now further confirmed that this lactate accumulation arised from a clear metabolic shift, where the upregulation of glycolysis is coupled with the downregulation of oxidative phosphorylation.”

“The raw non-targeted metabolomic data are available online through OMIX database (<https://ngdc.cnbc.ac.cn/omix/>) with the study accession ID: OMIX011916.”

Comment #3: If the authors want to discuss shifts between energy-supplying pathways, they should demonstrate that the actual transition from one to another occurs. What is the relative expression of genes involved in oxidative phosphorylation? Is glycolysis indeed the dominating pathway used for energy generation in the hypertrophic skin?

Reply:

Thank you for your insightful and constructive comments. We agree that demonstrating a clear shift between energy-supplying pathways is crucial for our manuscript. Your feedback has prompted us to perform additional experiments that we believe have substantially strengthened our conclusions. To directly address your questions, we have performed new RT-qPCR experiments to assess the relative mRNA expression of key genes involved in oxidative phosphorylation (OXPHOS) and the TCA cycle in hypertrophic scar tissues compared to normal skin tissues. We specifically analyzed *PDHAI* (a key component of the pyruvate

dehydrogenase complex, which links glycolysis to the TCA cycle), *CS*, *IDH1*, *IDH2*, and *OGDH* (critical enzymes within the TCA cycle). Our new results show a significant downregulation in the expression of *PDHA1*, *IDH1*, and *IDH2* in HS tissues. However, we observed a decreasing trend for *CS* and *OGDH*, although these did not reach statistical significance in our cohort. Furthermore, we have performed a non-targeted metabolomic analysis of HS and NS tissues. The results indicated that key intermediates of the TCA cycle, such as citric acid, were significantly decreased in HS. Conversely, metabolites of the glycolytic pathway, such as phosphoenolpyruvic acid, were significantly increased (Figure S1A-S1C).

This demonstrated downregulation of the OXPHOS pathway, when viewed in conjunction with our original findings, provides compelling evidence for a metabolic transition. Our original data already showed: (1) Significantly elevated lactate levels in HS tissues (Figure 1A). (2) Upregulated expression of key glycolysis-related genes and transporters (*PKM2*, *LDHA*, *MCT1*, *MCT4*) (Figure 1B-E). (3) A heightened extracellular acidification rate (ECAR) in macrophages, which are the primary source of lactate in the stiff HS microenvironment (Figure 2). The combination of these findings—the upregulation of glycolysis and the concurrent downregulation of OXPHOS—strongly supports the conclusion that a metabolic shift towards aerobic glycolysis occurs in hypertrophic scars. This shift establishes glycolysis as a dominant metabolic pathway fueling the profibrotic conditions, particularly by providing the lactate substrate for the epigenetic modifications that are central to our study.

We have revised the Results and Discussion sections of the manuscript to incorporate these new findings and to more explicitly articulate this metabolic transition. We believe these additions and revisions fully address your concerns and have made our manuscript much more robust. Thank you again for your valuable guidance.

Revised manuscript:

“To characterize the metabolic phenotype, non-targeted metabolomic analysis of HS and

healthy tissues revealed a distinct metabolic profile in patients (Figure S1A; Supplemental data 1). HS tissue exhibited a significant decrease in the key TCA cycle intermediate, citric acid, and a concomitant increase in the glycolytic metabolite, phosphoenolpyruvic acid (Figure S1A-S1C).”

“Next, RT-qPCR analysis of the expression of genes related to glycolysis (LDHA, HK2, PKM2 and PFKFB3), lactate transport (MCT1 and MCT4) and OXPHOS (PDHA1, CS, IDH1, IDH2 and OGDH) showed a significant increase of lactate-related genes in HS, particularly MCT1 (Figure 1B). Specifically, the mRNA levels of PDHA1, IDH1 and IDH2 were significantly decreased compared to normal skin (Figure 1B). This concurrent upregulation of the glycolytic pathway and downregulation of key OXPHOS components provides strong evidence of a metabolic switch towards aerobic glycolysis in the HS microenvironment.”

“We have now further confirmed that this lactate accumulation arised from a clear metabolic shift, where the upregulation of glycolysis is coupled with the downregulation of oxidative phosphorylation.”

Revised figures:

Figure 1D

Figure S1

Comment #4: The quality of IF is very poor, and it's difficult to see individual cells. Higher quality imaging should be provided.

Reply:

Thank you for your insightful and constructive comments. We agree that the quality of the original immunofluorescence images was insufficient and we apologize that this made it difficult to visualize individual cells clearly. To address this important issue, we have taken the following steps to improve the manuscript: (1) Increased image resolution: We have increased the resolution of all immunofluorescence images presented in the manuscript. This should provide much greater detail and clarity. (2) Image replacement: We have carefully reviewed all our IF images and have replaced several of the lower-quality ones with new, clearer images from our experiments (for example, in Figure 1E and 1F). In these new images, the morphology and outlines of individual cells are now clearly discernible. We have updated all relevant figures in the revised manuscript. We believe these changes have significantly improved the quality of our data presentation. We are grateful for the opportunity to improve our manuscript and hope the revised images are now satisfactory. Thank you again for your constructive suggestion.

Revised figures:

Figure 1E

Figure 1F

Comment #5: There is a discrepancy in the relative protein expression measured by Western blot, IHC, and IF in Figure 1. The authors should explain why.

Reply:

We sincerely thank the reviewer for their careful reading of our manuscript and for this insightful comment. We acknowledge the reviewer's observation regarding the apparent variations in the magnitude of relative protein expression for key glycolytic markers, particularly MCT1, as measured by WB, IHC, and IF in Figure 1. We agree that this point warrants clarification. The variations observed do not undermine our conclusion but rather stem from two main factors: the inherent biological variability among patient samples and the fundamental technical differences between these analytical methods.

The primary reason for this variation is the biological heterogeneity among human subjects. As detailed in our Methods section, the HS and NS samples were collected from a cohort of different patients. For ethical and practical reasons, the exact same tissue specimen from a single patient was not used across all three experimental procedures (WB, IHC, and IF). While we analyzed multiple biological replicates for each experiment (e.g., n=5 for WB, as stated in the Figure 1 legend) to ensure the statistical validity of the observed trend, patient-to-patient variability in protein expression levels is expected. Furthermore, it is crucial to recognize the distinct nature of these three techniques, each providing a different type of information: (1) Western blotting provides a semi-quantitative analysis of the average protein expression across a bulk population of cells from a homogenized tissue lysate. In contrast, IHC and IF are primarily qualitative or semi-quantitative methods that provide indispensable spatial information, revealing the cellular localization and distribution of the target protein within the intact tissue architecture. For instance, our IF co-staining in Figure 1F was crucial to demonstrate that MCT1 is specifically upregulated in α -SMA-positive myofibroblasts in HS tissue, a critical finding that a bulk lysate WB could not resolve. (2) The sample preparation for each method is vastly different. WB analysis involves harsh protein denaturation and solubilization from lysates. IHC and IF, however, rely on formalin-fixed, paraffin-embedded tissue sections that require antigen retrieval procedures to unmask

epitopes. These distinct processes can influence the antibody's binding affinity and, consequently, the resulting signal intensity. (3) The signal detection and amplification systems are unique to each method. WB uses an enzyme-conjugated secondary antibody that generates a chemiluminescent signal, quantified via densitometry. IHC employs an enzymatic reaction (DAB staining) visualized by bright-field microscopy, while IF uses fluorescently labeled antibodies visualized by fluorescence or confocal microscopy. These methods have different sensitivities, dynamic ranges, and signal-to-noise ratios, making a direct comparison of expression magnitudes across these platforms inappropriate.

In summary, while the exact magnitude of expression may differ, the key conclusion remains highly consistent and robust across all three independent assays: **the expression of MCT1 and the overall capacity for glycolysis are significantly elevated in hypertrophic scar tissue compared to normal skin.** The consistency of this directional trend across multiple methodologies strongly supports the validity of our findings.

To address the reviewer's excellent point and improve the manuscript's clarity, we will add a sentence to the Results section (in the paragraph corresponding to Figure 1) to briefly state that while all methods confirm the upregulation of these proteins in HS, a direct comparison of the magnitude of change is not appropriate due to inter-patient variability and the distinct technical nature of each assay. Once again, we thank the reviewer for their constructive feedback, which has helped us to enhance the precision of our manuscript.

Revised manuscript:

“Despite having some minor differences, similar patterns were observed in IHC assay and immunofluorescence results (Figure 1D and 1E), which indicated that the expression of MCT1 and the overall capacity for glycolysis are significantly elevated in HS tissue compared to NS.”

Comment #6: The authors talk about data that is not present in the figure: “ and we did not observe noticeable differences of MCT1 intensity in vascular endothelial cells between NS

and HS (Figure 1F)” – no markers of vascular endothelial cells has been used in the whole figure, and in particular not the referred Figure 1F.

Reply:

We sincerely thank you for your careful and detailed review of our manuscript and for pointing out this important inconsistency. We agree completely with the reviewer. Our original statement regarding the lack of change in MCT1 intensity in vascular endothelial cells was an overstatement and not directly supported by the data presented. The reviewer is correct that Figure 1F showed co-staining for MCT1 and α -SMA, which is a marker for myofibroblasts and smooth muscle cells, but not a specific marker for vascular endothelial cells. Our initial comment was based on a general morphological observation of the blood vessel structures, but it was not appropriate to make a definitive claim without using a specific endothelial marker (such as CD31). To address this issue and ensure the scientific accuracy of our manuscript, we have removed the unsubstantiated sentence from the revised text. The deleted sentence is: "and we did not observe obvious differences of MCT1 intensity in vascular endothelial cells between NS and HS (Figure 1F)". We appreciate the reviewer's valuable feedback, which has helped us improve the precision and quality of our paper.

Comment #7: In Figure 2, the authors use an in vitro system to investigate the cellular origin of lactate, which is detected in hypertrophic skin tissue. This approach is reductionistic, as it does not aim to identify which of the various cell types in the hypertrophic skin is the primary producer of lactate. Instead, they select specific cells to probe for lactate production. The authors should either use patient material (as in Figure 1) or the developed mouse model to answer this question.

Reply:

We thank you for this insightful and constructive comment. We agree that identifying the primary source of lactate directly within the complex cellular milieu of HS tissue is critical to substantiating our claims. The reviewer's concern regarding the reductionist nature of our

initial *in vitro* system is well-taken, and we recognize that relying solely on cell lines may not fully recapitulate the *in vivo* cellular dynamics. To address this pivotal point directly, we have performed three new sets of experiments using human patient-derived materials, as you suggested.

First, to provide direct quantitative evidence, we successfully isolated primary macrophages, endothelial cells, and fibroblasts using magnetic activated cell sorting (MACS) from both fresh HS and NS tissues. We then directly measured the extracellular and intracellular lactate levels in these distinct primary cell populations. We found that primary macrophages isolated from HS tissues exhibited the most substantial and significant increase in lactate content compared to their counterparts from NS when compared to other cell types isolated from HS (Figure 2G). In addition, enhanced ECAR level was observed in primary macrophages from HS tissues compared to control (Figure 2H). Second, we conducted multispectral immunofluorescence staining on human HS and NS tissues to investigate the *in situ* activity of the glycolytic pathway in different cell types. Our new results reveal a significant upregulation of key glycolytic enzymes (HK2 and LDHA) specifically within the macrophage population in HS tissues (Figure 2I and 2J). In contrast, endothelial cells did not show statistically significant changes in glycolytic activity between HS and NS (Figure S3D and S3E). This provides strong evidence from patient material that macrophages, in particular, adopt a hyper-glycolytic phenotype within the scar microenvironment.

Furthermore, we have performed a new *in vivo* experiment using our full-thickness skin defect mouse model. In this experiment, we selectively depleted macrophages during the wound healing process to directly assess their contribution to lactate production and subsequent scar formation. Specifically, we administered clodronate liposomes sub-eschar every three days starting on day 7 post-injury to deplete macrophages, with a control group receiving saline injections. On day 21, wound tissues were harvested for analysis.

Immunofluorescence staining for the macrophage marker F4/80 confirmed a significant reduction in macrophage numbers in the scar tissues of clodronate-treated mice compared to controls (Figure S3F). Crucially, lactate concentration was markedly decreased in the scar

tissue of macrophage-depleted mice (Figure S3J). Moreover, clodronate liposomes significantly reduced scar formation in the dorsal wounds of mice and promoted hair follicle regeneration (Figure S11D-F).

Together, these new findings, derived directly from patient tissue and mouse model, strongly corroborate the hypothesis generated from our *in vitro* system (the original Figure 2) and provide a more robust and clinically relevant answer to the question of the cellular origin of lactate. We have now incorporated these new data into a new figure in the manuscript and revised the Results section accordingly. We believe these additions significantly strengthen the manuscript by providing direct, patient-derived evidence that pinpoints macrophages as the major source of lactate in the HS microenvironment. We are grateful to the reviewer for prompting this important validation. Thank you again for your guidance.

Revised figures:

Figure 2G

Figure 2H

Figure 2I

Figure 2J

Figure S3

Figure S11D-S11F

Revised manuscript:

“To validate in vivo that macrophages are a primary source of lactate during scar formation, we isolated primary macrophages, vascular endothelial cells and dermal fibroblasts from fresh NS and HS tissues using magnetic activated cell sorting (MACS) and protein-level validation was performed on these cells using western blotting (Figure S3A-S3C). In line with previous results, primary macrophages from HS tissues showed a profoundly elevated extracellular lactate accumulation and intracellular production, but lactate levels in fibroblasts and endothelial cells remained constant (Figure 2G). In addition, enhanced ECAR level was observed in primary macrophages from HS tissues compared to control (Figure 2H). This macrophage-specific metabolic shift was corroborated by the selective upregulation of glycolytic enzymes HK2 and LDHA within CD68⁺ macrophages in HS tissue (Figure 2I and 2J), with no corresponding change in CD31⁺ endothelial cells (Figure S3D and S3E). Crucially, we utilized clodronate liposomes to specifically deplete macrophages in the full-thickness skin defect model on the dorsal region of C57BL/6 mice. Immunofluorescence

analysis confirmed a significant depletion of F4/80-positive macrophages in the scar tissue of clodronate liposomes-treated mice compared to saline-treated controls (Figure S3F).

Importantly, this macrophage depletion resulted in a marked decrease in lactate levels within the scar tissue (Figure S3G). Collectively, these *in vivo* findings strongly support our conclusion that macrophages are a key contributor to the lactate-rich microenvironment in HS.”

“To identify the source of lactate in the scar microenvironment *in vivo*, C57BL/6 mice aged 6 to 8 weeks were randomly assigned to two groups (Control, Clodronate liposome).

Clodronate liposomes (Yeasen, 40337ES) were administered of 40 μ L sub-eschar every three days starting from day 7 post-injury with a solution prepared at a dose of 50ug/ml.”

“To validate the role of *in vivo* macrophage depletion, reducing the source of lactate, in scar formation, we then performed H&E and Masson's trichrome staining on the collected scar tissue at post-injury day 21. The results indicated that clodronate liposomes significantly reduced scar formation in the dorsal wounds of mice and promoted hair follicle regeneration, suggesting that macrophages play an important pathological role in the late stage of wound healing (Figure S11D-F). Collectively, MCT1 inhibition via AZD3965 or clearance of the source of lactate may be an effective therapeutic strategy for hypertrophic scar.”

Comment #8: The molecular interaction modeling exercise doesn't contribute to the main message of the study.

Reply:

We sincerely thank you for their insightful feedback and for raising this important point. We appreciate the opportunity to clarify the essential role of the molecular interaction modeling in substantiating our study's central message. We concede that the manuscript may not have made this link sufficiently explicit, and we have revised the text to better integrate these findings.

Our study's main message is not only the identification of a linear pathway (macrophage lactate → MCT1 → H3K231a → HEY2/COL11A1 expression) but, more critically, the discovery of a **self-reinforcing positive feedback loop** that perpetuates the fibrotic state in hypertrophic scars (HS). The molecular modeling is the linchpin that provides the direct mechanistic evidence for this feedback loop. Here is a step-by-step explanation of its indispensability to our core argument: (1) Establishing the physical interaction: Our experimental data, including Co-immunoprecipitation (Figure 8B) and immunofluorescence colocalization assays (Figures 8C, 8D), provided strong evidence that a physical interaction occurs between COL11A1 (a downstream product of the pathway) and the lactate transporter MCT1 (a key upstream component). This observation was the first indication of a potential feedback mechanism. (2) The crucial question: what is the functional consequence of this interaction? While our biochemical and cellular data confirmed that COL11A1 and MCT1 bind, these experiments could not explain the functional consequence of this interaction. Does this binding event alter MCT1's activity? If so, how? Answering this question is paramount to proving that the pathway is not merely linear but circular and self-amplifying. Without understanding the functional outcome, the feedback loop remains a speculative concept. (3) The role of molecular modeling: bridging the gap from interaction to function. This is the precise knowledge gap that our molecular interaction modeling was designed to fill. Its contributions are twofold: First, the docking simulations provided a structurally plausible model for how these two proteins interact, identifying specific amino acid residues that likely mediate the binding (Figure 8E). This elevates our finding from a simple observation to a concrete, testable molecular mechanism. Second, and most critically, our molecular dynamics simulations and binding free energy calculations revealed a crucial functional consequence: the binding of COL11A1 not only stabilizes the conformation of MCT1 (Figure S10A, S10B) but also significantly enhances its affinity for lactate (Figure 8G and S10D). This computational result directly supports our experimental data showing that COL11A1 expression correlates with increased intracellular lactate levels (Figure S10D).

Therefore, the molecular modeling is not an ancillary exercise; it is the critical evidence that mechanistically closes the proposed feedback loop: lactate → H3K231a → COL11A1 →

COL11A1 binds MCT1 → MCT1 affinity for lactate increases → more lactate enters the cell, thus perpetuating the cycle. Without this modeling, we could only state that COL11A1 and MCT1 interact. With the modeling, we can propose a complete, mechanistically detailed narrative: that the upregulation of COL11A1 in HS is not just a marker of fibrosis but an active driver that enhances the metabolic reprogramming that initiated its expression. This transforms our understanding of HS pathogenesis from a linear process to a self-sustaining disease state, which we believe is the most significant message of our study. We hope this explanation clarifies the indispensable role of the modeling data. We have revised the relevant sections of the manuscript (particularly in the Results and Discussion) to state more clearly how the modeling provides the functional explanation for the positive feedback loop. Thank you again for your valuable comment, which has helped us to improve the clarity and impact of our paper.

Revised manuscript:

“While our Co-IP and immunofluorescence data confirmed a physical interaction between COL11A1 and MCT1, the functional consequence of this binding remained unclear. To determine precisely how COL11A1 binding might modulate MCT1's activity, we performed molecular modeling.”

“The simulations provide a structural and energetic basis for how COL11A1 enhances MCT1's lactate transport affinity, thereby closing the loop.”

“Furthermore, the COL11A1 protein exhibits a non-canonical function by directly interacting with the lactate transporter MCT1. Molecular modeling shows this interaction stabilizes MCT1 and enhances its lactate affinity, potentiating lactate transport. This establishes a pathogenic, self-reinforcing positive feedback loop (lactate-H3K231a-COL11A1-MCT1), which sustains the scar microenvironment.”

Reviewer 4:

In the study titled “Lactate derived from macrophages drives skin dermal fibroblasts phenotypic remodeling via MCT1-primed histone H3 lysine 23 lactylation in hypertrophic scar,” Hu et al. delineated the role of macrophage-derived lactate in hypertrophic scar (HS) formation. They identified that macrophage-derived lactate serves as a critical mediator of fibroblast phenotypic remodeling through monocarboxylate transporter 1 (MCT1)-mediated histone H3 lysine 23 lactylation (H3K23la) in HS. The authors observed that fibroblast-specific deletion of Mct1 or pharmacological inhibition of Mct1 in mice reduced collagen deposition, accelerated wound healing, and attenuated scar formation.

The study is highly interesting, well designed, and clearly presented. Nevertheless, several minor issues need to be addressed:

Comment #1: Multiple immunofluorescence images in the manuscript are of relatively low resolution (e.g., Figure 1E and Figure S6). Higher-resolution images should be provided.

Reply:

Thank you for your valuable feedback and for your careful review of our manuscript. We sincerely appreciate your constructive comments. We agree that the clarity and resolution of images are critical for presenting our findings accurately, and we thank you for bringing this issue to our attention.

In response to your concern, we have taken immediate action to improve the quality of the figures in our manuscript. We have replaced all the immunofluorescence images that were of low resolution, including the specific examples you mentioned, Figure 1E and Figure S9, with new, high-resolution versions. The issue in the original submission was likely due to an unintentional loss of quality during the file conversion and PDF generation process. To address this thoroughly, we have returned to our original, raw image files captured by the microscope. We have carefully re-exported these images at a high resolution and have re-assembled the figures to ensure that all details are sharp and clearly visible. Furthermore, we

have taken this opportunity to conduct a comprehensive review of all figures throughout the manuscript and its supplementary materials to ensure that they all meet the highest standards for publication. We are confident that the revised images in the updated manuscript now clearly and accurately illustrate our data. We hope these improvements satisfactorily address your concerns. Thank you once again for your time and guidance.

Revised figures:

Figure 1E

Figure S9F

Figure S9G

Comment #2: In Figure 9C and Figure 10C, the blue coloration of the Masson's trichrome staining is uneven, suggesting possible overstaining. Higher-resolution Masson's staining images are required.

Reply:

We sincerely thank the reviewer for their careful evaluation and valuable feedback on our manuscript. We agree that high-quality images are crucial for conveying our findings

accurately.

In response to the reviewer's comments, we have taken the following actions: (1)

Replacement of images with higher-resolution versions: As requested, we have replaced the original Masson's trichrome staining images in Figure 9D and Figure 10C with new, higher-resolution versions. We believe these updated images provide significantly greater detail and clarity, allowing for a more thorough assessment of the tissue histology. These changes have been implemented in the revised manuscript. (2) Clarification on the "uneven" blue coloration and "overstaining": We appreciate the reviewer pointing out the "uneven" and intense blue coloration in certain panels. We would like to respectfully clarify that this observation is, in fact, a key histopathological finding of our study and is not an artifact of overstaining. In our study, Masson's trichrome staining is used to visualize collagen deposition, a hallmark of fibrosis and hypertrophic scar. The key characteristics of hypertrophic scar are the excessive synthesis and disorganized, dense deposition of collagen fibers. In the control and lactate-treated groups (e.g., the *Mct1^{fl/fl}Colla2-cre-* + NaLac group in Fig. 9D and the Control + NaLac group in Fig. 10C), the tissue exhibits significant fibrosis. This pathological state is characterized by thick, irregularly arranged, and densely packed collagen bundles. Masson's stain, which stains collagen blue, therefore produces an intense, deep, and non-uniform (uneven) blue appearance that accurately reflects this disorganized and excessive collagen matrix. Conversely, in the groups where scar formation was ameliorated through our interventions (i.e., the *Mct1* knockout group in Fig. 9D and the AZD3965-treated group in Fig. 10C), the wound healing process is improved. This results in collagen fibers that are finer, more organized, and less dense, arranging themselves in a smoother, parallel fashion that more closely resembles normal dermis. Consequently, the blue staining in these panels appears lighter and more uniform, which is indicative of reduced fibrosis. Therefore, the variation in staining intensity and the "unevenness" observed across the different experimental groups are a direct visual representation of the biological differences in collagen architecture and density, which supports our central conclusion that inhibiting the MCT1-lactate axis ameliorates hypertrophic scar formation.

We are confident that the new high-resolution images, combined with this explanation, will make this crucial distinction even clearer to the reader. We thank the reviewer again for providing us with the opportunity to clarify this important point and improve the quality of our figures.

Figure 9D

Figure 10C

Comment #3: The authors demonstrated *in vitro* that macrophages are the primary source of lactate accumulation in the scar metabolic microenvironment. Given the substantial differences between *in vitro* and *in vivo* conditions, *in vivo* validation of macrophage-mediated lactate accumulation in the hypertrophic scar microenvironment would further strengthen the credibility of this conclusion.

Reply:

Thank you for your constructive and insightful feedback on our manuscript. We agree that validating our *in vitro* findings *in vivo* is crucial for strengthening the conclusions of our study. Your point regarding the substantial differences between *in vitro* and *in vivo* conditions is well-taken, and we appreciate the opportunity to enhance the credibility of our work.

To address this important concern, we have performed three additional sets of experiments using human hypertrophic scar (HS) and normal skin (NS) tissues to validate our conclusion that macrophages are the primary source of lactate in the HS microenvironment. First, to provide more direct and quantitative evidence, **we successfully isolated primary macrophages, endothelial cells, and fibroblasts from freshly obtained human NS and HS tissues using magnetic activated cell sorting (MACS). We then directly measured the intracellular lactate content of these distinct cell populations.** Consistent with our *in vitro* data (original Figure 2), we found that primary macrophages isolated from HS tissues exhibited the most significant increase in lactate levels compared to those from NS (Figure 2G). This increase was markedly more pronounced than that observed in fibroblasts or endothelial cells from the same HS tissues (Figure 2G). In addition, enhanced ECAR level was observed in primary macrophages from HS tissues compared to control (Figure 2H). Second, we conducted multicolor immunofluorescence staining on human HS and NS specimens. **We co-stained for macrophage markers (CD68), endothelial cell markers (CD31), and key glycolytic enzymes (e.g., HK2 and LDHA).** The results clearly demonstrated a significant upregulation of these glycolytic enzymes specifically within the CD68⁺ macrophage population in HS tissues (Figure 2I and 2J). In contrast, no statistically significant changes were observed in the CD31⁺ endothelial cells (Figure S3D and S3E). This provides strong *in vivo* evidence that macrophages in the scar microenvironment possess a heightened glycolytic state compared to other cell types. Furthermore, we have performed a new *in vivo* experiment using our full-thickness skin defect mouse model. **In this experiment, we selectively depleted macrophages during the wound healing process to directly assess their contribution to lactate production.** Specifically, we administered clodronate liposomes subcutaneously every three days starting on day 7 post-injury to deplete macrophages, with a control group receiving saline injections. On day 21, wound tissues were harvested for analysis. Immunofluorescence staining for the macrophage marker F4/80 confirmed a significant reduction in macrophage numbers in the scar tissues of clodronate-treated mice compared to controls (Figure S3F). Crucially, lactate concentration was markedly decreased in the scar tissue of macrophage-depleted mice (Figure S3G). Moreover, clodronate liposomes significantly reduced scar formation in the dorsal wounds of mice and

promoted hair follicle regeneration (Figure S11D-F).

Together, these new findings, derived directly from human clinical samples and mouse model, robustly corroborate our original conclusion based on the *in vitro* model. They provide compelling evidence that macrophages are indeed a major source of lactate accumulation within the hypertrophic scar metabolic microenvironment. We have incorporated these new results into the revised manuscript. We have added a new Figure 2G-2J and S3 to present the data from these experiments and have updated the Results section. We have also added detailed protocols for these new experiments to the Methods section. We are grateful for your valuable suggestion, which has allowed us to substantially strengthen our manuscript. We hope that these additions and revisions have fully addressed your concerns. Thank you once again for your time and guidance.

Revised figures:

Figure 2G

Figure 2H

Figure 2I

Figure 2J

Figure S3

Figure S11D-S11F

Revised manuscript:

“To validate *in vivo* that macrophages are a primary source of lactate during scar formation, we isolated primary macrophages, vascular endothelial cells and dermal fibroblasts from fresh NS and HS tissues using magnetic activated cell sorting (MACS) and protein-level validation was performed on these cells using western blotting (Figure S3A-S3C). In line with previous results, primary macrophages from HS tissues showed a profoundly elevated extracellular lactate accumulation and intracellular production, but lactate levels in fibroblasts and endothelial cells remained constant (Figure 2G). In addition, enhanced ECAR level was observed in primary macrophages from HS tissues compared to control (Figure 2H). This macrophage-specific metabolic shift was corroborated by the selective upregulation of glycolytic enzymes HK2 and LDHA within CD68⁺ macrophages in HS tissue (Figure 2I and 2J), with no corresponding change in CD31⁺ endothelial cells (Figure S3D and S3E). Crucially, we utilized clodronate liposomes to specifically deplete macrophages in the full-thickness skin defect model on the dorsal region of C57BL/6 mice. Immunofluorescence

analysis confirmed a significant depletion of F4/80-positive macrophages in the scar tissue of clodronate liposomes-treated mice compared to saline-treated controls (Figure S3F). Importantly, this macrophage depletion resulted in a marked decrease in lactate levels within the scar tissue (Figure S3G). Collectively, these *in vivo* findings strongly support our conclusion that macrophages are a key contributor to the lactate-rich microenvironment in HS.”

“To identify the source of lactate in the scar microenvironment *in vivo*, C57BL/6 mice aged 6 to 8 weeks were randomly assigned to two groups (Control, Clodronate liposome). Clodronate liposomes (Yeasen, 40337ES) were administered of 40 μ L sub-eschar every three days starting from day 7 post-injury with a solution prepared at a dose of 50ug/ml.”

“To validate the role of *in vivo* macrophage depletion, reducing the source of lactate, in scar formation, we then performed H&E and Masson's trichrome staining on the collected scar tissue at post-injury day 21. The results indicated that clodronate liposomes significantly reduced scar formation in the dorsal wounds of mice and promoted hair follicle regeneration, suggesting that macrophages play an important pathological role in the late stage of wound healing (Figure S11D-F). Collectively, MCT1 inhibition via AZD3965 or clearance of the source of lactate may be an effective therapeutic strategy for hypertrophic scar.”

Comment #4: The authors identified an interaction between COL11A1 and MCT1. However, it remains unclear whether additional proteins are involved in this interaction. Further experimental validation or detailed discussion in the Discussion section is warranted.

Reply:

Thank you for your insightful and constructive comment. We agree that this is a crucial point that warrants further consideration. Our Co-IP and colocalization results strongly indicate a robust interaction between COL11A1 and MCT1 within the cellular context. Furthermore, our computational modeling supports the plausibility of a direct binding interface.

However, we concur that in a complex biological system, this interaction is likely not occurring in isolation and may be stabilized or mediated by other associated proteins. Your query prompted us to delve deeper into the literature and consider potential candidates. A particularly strong candidate for such a role is **CD147 (also known as Basigin or EMMPRIN)**. It is well-established that CD147 acts as an essential ancillary/chaperone protein for the proper trafficking, cell surface expression, and functional activity of monocarboxylate transporters, including MCT1 (Nan Wang et al., 2021; Andrew P Halestrap, 2013). Without association with CD147, MCT1 is often retained intracellularly and rapidly degraded. Given that our study focuses on the function of MCT1 at the plasma membrane for lactate transport, the involvement of CD147 is almost certain and highly relevant. Moreover, CD147 itself is deeply implicated in ECM remodeling and fibrosis. As an "Extracellular Matrix Metalloproteinase Inducer" (EMMPRIN), it stimulates the production of MMPs, which are critical for tissue remodeling in processes like wound healing and scarring. Its expression is often upregulated in fibrotic conditions and it plays a significant role in cell-matrix interactions (Jiao Wu et al., 2022; Mingchuan Liu et al., 2023). Therefore, we propose a compelling hypothesis that the interaction we observed may exist as a **ternary complex of COL11A1-CD147-MCT1**. In this model: (1) CD147 acts as the direct chaperone, binding to MCT1 to ensure its stability and presence at the cell surface. (2) COL11A1, as a key component of the fibrotic ECM, could then interact with the CD147-MCT1 complex. This interaction might serve to anchor the lactate transport machinery within specific membrane domains, or it could sterically stabilize the complex, thereby enhancing MCT1-mediated lactate transport and perpetuating the positive feedback loop we described. While definitively validating this ternary complex through further experiments like sequential Co-IP is beyond the scope of our current study, we believe this is a highly plausible mechanism and a fascinating avenue for future research.

To address your excellent point, we have now added a detailed paragraph in the Discussion section to elaborate on the potential role of CD147 as a key intermediary in the COL11A1-MCT1 interaction, citing relevant literature. We believe this addition significantly enriches the manuscript and provides a more complete mechanistic picture for our readers. Thank you

again for helping us improve the quality of our manuscript.

Revised manuscript:

“Furthermore, while our co-immunoprecipitation and computational data support an interaction between COL11A1 and MCT1, it is critical to consider the potential involvement of other proteins that may mediate or stabilize this complex in the cellular milieu. A prime candidate is the chaperone protein CD147, which is indispensable for the membrane localization and functional stability of MCT1. We thus hypothesize that COL11A1 forms a ternary complex with CD147 and MCT1. This complex would anchor the lactate transporter, enhancing its activity and thereby amplifying pro-fibrotic signaling. Future sequential immunoprecipitation experiments are required to elucidate the precise composition and stoichiometry of this putative complex.”

References:

- (1) Wang, N., Jiang, X., Zhang, S., Zhu, A., Yuan, Y., Xu, H., Lei, J., & Yan, C. (2021). Structural basis of human monocarboxylate transporter 1 inhibition by anti-cancer drug candidates. *Cell*, 184(2), 370–383.e13.
 - (2) Halestrap A. P. (2013). The SLC16 gene family - structure, role and regulation in health and disease. *Molecular aspects of medicine*, 34(2-3), 337–349.
 - (3) Wu, J., Chen, L., Qin, C., Huo, F., Liang, X., Yang, X., Zhang, K., Lin, P., Liu, J., Feng, Z., Zhou, J., Pei, Z., Wang, Y., Sun, X. X., Wang, K., Geng, J., Zheng, Z., Fu, X., Liu, M., Wang, Q., Chen, Z. N. (2022). CD147 contributes to SARS-CoV-2-induced pulmonary fibrosis. *Signal transduction and targeted therapy*, 7(1), 382.
 - (4) Liu, M., Peng, T., Hu, L., Wang, M., Guo, D., Qi, B., Ren, G., Wang, D., Li, Y., Song, L., Hu, J., & Li, Y. (2023). N-glycosylation-mediated CD147 accumulation induces cardiac fibrosis in the diabetic heart through ALK5 activation. *International journal of biological sciences*, 19(1), 137–155.
-

Comment #5: The authors observed elevated lactate levels and increased MCT1 expression in HS tissues. What is the relationship between lactate levels, MCT1 expression, and HS disease progression?

Reply:

Thank you for your insightful question, which highlights the central mechanism we aimed to elucidate in our study. The relationship between elevated lactate, increased MCT1 expression, and the progression of HS is indeed the cornerstone of our findings. This relationship is not merely a correlation but rather a causative, self-reinforcing feedback loop that drives the fibrotic process. We have detailed this mechanism throughout our manuscript, and we appreciate the opportunity to summarize and clarify it here.

Based on our findings, the relationship can be understood through the following sequential and cyclical steps: (1) Origin of lactate: We identified that in the stiff, pathological microenvironment of HS, macrophages undergo a metabolic shift to a hyperglycolytic phenotype. These activated macrophages become the primary source of the elevated lactate levels observed in HS tissues (as shown in our findings related to Figure 2 and S3). (2) Lactate uptake by fibroblasts via MCT1: Concurrently, we observed a significant upregulation of MCT1 expression specifically on dermal fibroblasts, particularly myofibroblasts, within the HS tissue (Figure 1). This increased MCT1 expression functions as a crucial gateway, facilitating the transport of the abundant macrophage-derived lactate from the extracellular space into the fibroblasts. (3) Lactate as an epigenetic signal: Once inside the fibroblast, lactate is not merely a metabolic byproduct but acts as a substrate for a key epigenetic modification—histone lactylation. We demonstrated that this lactate influx leads to a specific increase in H3K231a in the myofibroblasts of HS (Figure 4). (4) Activation of a profibrotic program: The elevated H3K231a then acts as an epigenetic activator, promoting the transcription of key profibrotic genes, namely HEY2 and COL11A1 (Figure 5). These genes are critical drivers of the fibroblast-to-myofibroblast transition and extracellular matrix deposition, which are hallmarks of HS progression. (5) A vicious positive feedback loop driving progression: This is the most critical part of the relationship, explaining the "disease

progression." The process becomes a self-perpetuating cycle: The upregulation of *HEY2* further amplifies fibrotic signaling through the YAP1/SMAD2 pathway (Figure 7). Crucially, the other upregulated protein, COL11A1, directly interacts with and stabilizes the MCT1 transporter on the fibroblast membrane (Figure 8). This stabilization enhances MCT1's efficiency, leading to even greater lactate uptake by the fibroblast. This creates a vicious positive feedback loop: (Extracellular lactate) → (MCT1-mediated influx) → (Increased H3K231a) → (Upregulated COL11A1) → (Stabilized, more efficient MCT1) → (Even more lactate influx). This self-reinforcing loop explains how the fibrotic state in HS is not only initiated but also maintained and amplified over time, thus driving disease progression. Our *in vivo* experiments, where genetic ablation or pharmacological inhibition of MCT1 led to attenuated scar formation and accelerated wound healing (Figures 9 and 10), provide strong evidence for the essential role of this axis in HS pathogenesis. The concept of lactate fueling pathological states is increasingly recognized in other diseases such as cancer and liver fibrosis (Hyunsoo Rho et al., 2023; George A Brooks et al., 2018).

We hope this detailed explanation fully addresses your question. We agree that this central relationship could be highlighted more explicitly in the manuscript. As per your suggestion, we will revise the Discussion section to include a more focused and clearer paragraph summarizing this metabolic-epigenetic feedback loop. Thank you again for your valuable feedback.

Revised manuscript:

“Our study establishes a novel axis linking metabolic reprogramming and epigenetic regulation in hypertrophic scar pathogenesis. First and foremost, we redefine the upstream trigger of fibroblast activation by identifying mechanically stressed macrophages as the key lactate producers. The stiff microenvironment induces macrophages to adopt a hyperglycolytic phenotype, producing lactate that is shuttled into fibroblasts via MCT1, which has a significant upregulation in hypertrophic scar tissue. Second, this lactate influx primes H3K231a, activating profibrotic genes *HEY2* and *COL11A1*. *HEY2* amplifies fibrotic signaling via YAP1/SMAD2, while *COL11A1* stabilizes MCT1 to enhance lactate transport,

creating a self-reinforcing loop that perpetuates fibrosis. Furthermore, the process initiates in the stiff HS microenvironment, which induces macrophages to produce excessive lactate. Critically, targeting this pathway via fibroblast-specific Mct1 deletion or pharmacological inhibition (AZD3965) mitigated the core pathology. This intervention resulted in two concomitant benefits: an acceleration of wound closure, likely reflecting a more rapid resolution of the pro-fibrotic inflammatory phase and a reduction in the excessive collagen deposition that defines the quality of the scar. This work provides the first evidence that macrophage-derived lactate fuels HS progression through MCT1-dependent histone lactylation, redefining lactate as a metabolic-epigenetic mediator rather than a passive byproduct. These findings position MCT1 as a therapeutic target and establish histone lactylation as a novel epigenetic driver of HS progression.”

References:

(1) Rho, H., Terry, A. R., Chronis, C., & Hay, N. (2023). Hexokinase 2-mediated gene expression via histone lactylation is required for hepatic stellate cell activation and liver fibrosis. *Cell metabolism*, 35(8), 1406–1423.e8.

(2) Brooks G. A. (2018). The Science and Translation of Lactate Shuttle Theory. *Cell metabolism*, 27(4), 757–785.

Comment #6: In Figure S6C, nonspecific bands are present for COL11A1, and the two bands in the si-NC group are difficult to distinguish. A clearer Western blot image should be provided.

Reply:

Thank you for your valuable feedback and the opportunity to revise our manuscript. We appreciate the thoroughness of your review, which has helped us improve the quality of our paper. We agree that the original Western blot image for COL11A1 in Figure S6C lacked the necessary clarity, with the presence of non-specific bands and difficulty in distinguishing the

bands in the control group. To address this concern, we have repeated the Western blot experiment for COL11A1 knockdown. The new experiment was performed with optimized conditions to reduce background and non-specific binding, resulting in a much clearer image with distinct, specific bands. This new result more effectively demonstrates the successful knockdown of COL11A1 following siRNA transfection. The improved, clearer Western blot image has now replaced the previous one in the revised Figure S9E of the supplementary materials. We believe this revision fully addresses the reviewer's concern and provides clear evidence for our findings. Thank you again for your valuable feedback.

Figure S9E

Response to Reviewers

Dear Editors and Reviewers,

Thank you for your letter and for the reviewers' comments concerning our manuscript entitled "Lactate derived from macrophages drives skin dermal fibroblasts phenotypic remodeling via MCT1-primed histone H3 lysine 23 lactylation in hypertrophic scar" (Manuscript Number: NCOMMS-25-39200). Those comments are all valuable and very helpful for revising and improving our manuscript, as well as the important guiding significance to our researches. We have studied comments carefully and have made corrections which we hope meet with approval. Revised portions are marked in yellow in the revised manuscript. The main corrections in the paper and the responses to the reviewer's comments are as following:

Responses to the reviewer's comments:

Reviewer #3 (Remarks to the Author):

It's appreciated that the authors have performed additional experiments in response to the previous review. These efforts have improved the manuscript and clarified several aspects of the study. Nonetheless, several important points remain outstanding, and further clarification and supporting evidence are needed to validate the study's conclusions fully:

Regarding the novelty of the findings

The authors have provided a detailed explanation of their view on the novelty of the study and have clarified how their findings integrate previously known concepts into a more specific mechanistic framework. While these additions and contextual revisions are appreciated, this reviewer remains unconvinced that the presented work constitutes a substantial conceptual advance. The roles of MCT1-mediated lactate transport, macrophage-derived lactate signaling, and histone lactylation in fibroblast activation have been previously established across several fibrotic contexts, including keloid and dermal fibrosis. The current study extends these findings to hypertrophic scar and delineates their interconnection in this specific model, but this represents more of a contextual refinement than a discovery of a

fundamentally new mechanism. While the addition of new molecular players (e.g., H3K231a, HEY2, and COL11A1) provides valuable mechanistic depth, the overarching concept of lactate-mediated metabolic–epigenetic crosstalk driving fibrosis remains consistent with prior reports. Thus, the study represents a meaningful refinement and contextualization of established mechanisms rather than the identification of a fundamentally new pathway.

Reply:

We sincerely thank the reviewer for this thoughtful and critical assessment. We fully understand your reservation regarding the conceptual novelty, given that lactate, MCT1, and histone lactylation have indeed been implicated in various fibrotic contexts, particularly in keloids and organ fibrosis. However, we respectfully submit that our study goes beyond a "contextual refinement." Based on the unique metabolic landscape of HS revealed in our data—which fundamentally differs from the established "Warburg effect" model in keloids—we believe our findings represent a distinct mechanometabolic mechanism. We wish to highlight three specific aspects where our work provides a substantial conceptual advance:

1. Redefining the Metabolic Landscape: HS is biologically distinct from Keloids

The reviewer correctly notes that lactate signaling is known in keloids. However, the source and the cellular logic of this lactate are fundamentally different in our HS model, representing a new paradigm for skin fibrosis. As we discuss in the revised text, previous literature establishes that keloid fibroblasts (KFs) themselves undergo metabolic reprogramming (aerobic glycolysis), exhibiting cancer-like bioenergetics with intrinsic lactate overproduction (Wang et al., 2021). In keloids, the fibroblast is the producer and the consumer (autocrine signaling). In sharp contrast, our targeted metabolomics and Seahorse assays (Figure 2) demonstrate that **HS fibroblasts do not exhibit glycolytic elevation** (consistent with Su et al., 2022). Instead, we identified that **mechanically stressed macrophages** are the primary lactate source. This establishes a "parasitic" or "coupled" metabolic circuit where fibroblasts rely on macrophage-derived lactate via MCT1. This distinction is crucial because it implies that targeting fibroblast glycolysis (a strategy

proposed for keloids) would fail in HS. Instead, blocking the transport (MCT1) or the source (macrophage stiffness-response) is required. This redefines the metabolic etiology of HS as distinct from keloids.

2. Identification of a Novel "Non-Canonical" Protein Interaction (COL11A1-MCT1 Loop)

While the general concept of "lactate leading to gene expression" is known, we discovered a fundamentally new molecular mechanism regarding how this signal is sustained. We found that the lactylation-induced gene COL11A1 does not merely function as an extracellular matrix structural protein. Our data (Figure 8 and molecular docking) reveals a **non-canonical intracellular function**: COL11A1 physically interacts with and stabilizes the MCT1 transporter on the plasma membrane. This creates a specific **positive feedback loop** (Macrophage Lactate→ Fibroblast MCT1→ H3K23la→ COL11A1→ Stabilized MCT1→ More Lactate influx). This self-reinforcing protein-protein interaction has not been reported in other fibrotic or cancer contexts and provides a structural explanation for the chronicity of the scar.

3. Specificity of the Epigenetic Code (H3K23la vs. H3K18la)

Previous studies in keloids identified H3K18la as the driver of fibrosis (Li et al., 2023). In contrast, our results identified H3K23la as the specific mark in HS myofibroblasts, while H3K18la levels remained unchanged (Figure 4B). This suggests that histone lactylation is not a generic "on-switch" for fibrosis but follows a specific "code" depending on the disease pathology (Keloid vs. HS). We mapped this specific H3K23la mark to the promoter of HEY2, a transcription factor not previously linked to this metabolic axis, which subsequently activates the YAP1/SMAD2 pathway. This delineates a precise, novel signaling axis (Lactate-H3K23la-HEY2-YAP1) rather than a generic application of lactylation.

In summary, while the components (lactate, MCT1) are known, the circuit topology we describe is novel and specific to HS: (1) **Input**: Macrophage mechano-transduction (not

fibroblast glycolysis). (2) **Mechanism:** A specific epigenetic mark (H3K231a) and a novel protein-stabilization loop (COL11A1-MCT1). (3) **Outcome:** A distinct therapeutic vulnerability (MCT1 inhibition in oxidative fibroblasts). We believe these findings provide the "mechanistic depth" and "differentiation from keloid pathophysiology" necessary to constitute a conceptual advance. We have taken your constructive feedback seriously and have thoroughly revised the manuscript to improve its clarity, accuracy, and impact. We thank you again for your valuable time and consideration.

Reference:

(1) Wang, Q., Wang, P., Qin, Z., Yang, X., Pan, B., Nie, F., & Bi, H. (2021). Altered glucose metabolism and cell function in keloid fibroblasts under hypoxia. *Redox biology*, 38, 101815.

(2) Su, Z., Jiao, H., Fan, J., Liu, L., Tian, J., Gan, C., Yang, Z., Zhang, T., & Chen, Y. (2022). Warburg effect in keloids: A unique feature different from other types of scars. *Burns : journal of the International Society for Burn Injuries*, 48(1), 176–183.

Original Comment 1:

Using in vitro model systems to dissect mechanistic details is acceptable when grounded in initial observations derived from the native tissue context. However, in this study, the focus on fibroblasts, macrophages, and endothelial cells appears selective rather than empirically established. A more systematic analysis that includes other key skin-resident cell types—such as adipocytes, mast cells, and additional immune populations—would have been essential to accurately identify the principal contributors to the observed interactions. Moreover, the use of cell lines for in vitro experiments represents a significant limitation, as these models often diverge from primary cells and therefore cannot faithfully reproduce the complexity of tissue-level signaling and cell–cell communication. This methodological choice reduces the physiological relevance of the mechanistic findings and may limit the generalizability of the conclusions to in vivo or clinical settings.

Reply:

We sincerely appreciate your insightful comments regarding the cellular complexity of the skin microenvironment and the methodological considerations of our in vitro models. We agree that a systematic analysis of other resident cell types is crucial to accurately identify the principal metabolic contributors. To address your concerns, we have performed additional experiments and included a detailed discussion based on both our new data and existing literature.

1. Regarding the selectivity of cell types and the inclusion of other key skin-resident cells:

We acknowledge that the skin environment is multicellular and that other cell types, such as mast cells, T cells, and adipocytes, play roles in wound healing. To determine if these cells contribute significantly to the lactate-rich microenvironment in HS, we conducted the following additional analyses: **(1) Mast cells and T cells:** We performed double immunofluorescence staining on human HS and NS tissues using markers for Mast cells

(Tryptase) and T cells (CD3) alongside key glycolytic enzymes (HK2 and LDHA). As shown in the newly added Figure S4G-S4I, we observed no statistically significant difference in the expression of HK2 or LDHA within Tryptase⁺ Mast cells or CD3⁺ T cells when comparing HS to NS tissues. While these immune cells are present in the scar tissue, our data suggests they do not undergo the profound "glycolytic switch" (upregulation of lactate-generating enzymes) observed in macrophages (Figure 2G-J). This reinforces our finding that macrophages are the predominant source of elevated lactate in the HS microenvironment. **(2)**

Adipocytes: Regarding adipocytes, we carefully considered their potential role based on tissue architecture and literature: Histologically, HS is characterized by excessive collagen deposition primarily in the reticular dermis, which significantly thickens and creates a spatial barrier separating the active fibrotic core from the subcutaneous adipose tissue (Marc G Jeschke et al., 2023). This physical distance suggests that paracrine metabolic signals (like lactate) from subcutaneous adipocytes are less likely to be the primary drivers of fibroblast phenotypes in the upper/mid-dermis compared to infiltrating macrophages which are in direct physical contact with fibroblasts. Current literature suggests the primary contribution of adipocytes to scarring is not through lactate secretion, but rather through the Adipocyte-Myofibroblast Transition (AMT), where intradermal adipocytes lose their lipid content and transdifferentiate into myofibroblasts (Brett A Shook et al., 2019). Therefore, in the context of our study—which focuses on the source of lactate driving epigenetic changes—adipocytes are less likely to be the "producers" and more likely to be a lineage responding to the fibrotic signals.

2. Regarding the use of cell lines vs. primary cells:

We appreciate your critique regarding the physiological relevance of cell lines. We wish to clarify that while cell lines were used for initial screening due to their stability in mechanical stiffness models, the core findings of our study were rigorously validated using primary cells and human tissues: **(1) Primary lactate source validation:** As shown in Figure 2G-J, we isolated primary macrophages, endothelial cells, and dermal fibroblasts directly from fresh patient tissues (NS and HS) using magnetic-activated cell sorting (MACS). We measured

intracellular/extracellular lactate levels and performed Seahorse metabolic flux analysis (ECAR and ATP production rate) on these primary cells, confirming that primary HS macrophages exhibit the highest glycolytic capacity and lactate secretion, consistent with our cell line data. **(2) Fibrosis mechanism validation:** For the downstream mechanistic experiments (histone lactylation, gene regulation), we primarily utilized **primary Human Skin Fibroblasts (HSFs and NSFs)** isolated from patient samples (Figures 1, 3, 4, 6, 7, 8) rather than cell lines. All key observations were further corroborated in human HS tissue samples via Western Blot and IHC (Figures 1, 4, 5) and in our *in vivo* mouse models.

We believe these additional data and clarifications demonstrate that macrophages are indeed the principal contributors to the lactate-rich microenvironment and that our findings are grounded in physiologically relevant primary cell and tissue contexts and we have revised the Result sections of our manuscript. We are grateful for your constructive criticism, which has significantly improved the quality and clarity of our manuscript. Thank you again for helping us strengthen the rigor of our study.

Revised figures:

Figure S4G

Figure S4H

Figure S4I

Figure S4J

Revised manuscript:

“To further rule out the contribution of other immune cell populations to the lactate-rich microenvironment, we also performed immunofluorescence co-staining of key glycolytic enzymes (HK2 or LDHA) with markers for mast cells (Tryptase) and T cells (CD3) in human NS and HS tissues. Quantitative analysis revealed no statistically significant differences in the number of glycolytic-enzyme-positive mast cells or T cells (Figure S4G-S4J).”

“Although the skin comprises a heterogeneous population of cells, our systematic analysis identifies macrophages as the principal source of lactate.”

“While adipocytes are key components of the skin stroma, their contribution to the lactate pool in HS was considered less prominent due to spatial constraints; the significantly thickened reticular dermis in HS creates a physical barrier that separates the fibrotic core from the subcutaneous adipose layer, limiting the potential for adipocyte-derived paracrine metabolic signaling to reach dermal fibroblasts. Additionally, our supplementary data confirmed that vascular endothelial cells and other resident immune cells, such as mast cells and T cells, do not exhibit the robust glycolytic upregulation observed in macrophages.”

References:

(1) Jeschke, M. G., Wood, F. M., Middelkoop, E., Bayat, A., Teot, L., Ogawa, R., & Gauglitz, G. G. (2023). Scars. *Nature reviews. Disease primers*, *9*(1), 64.

(2) Shook, B. A., Wasko, R. R., Rivera-Gonzalez, G. C., Salazar-Gatzimas, E., López-Giráldez, F., Dash, B. C., Muñoz-Rojas, A. R., Aultman, K. D., Zwick, R. K., Lei, V., Arbiser, J. L., Miller-Jensen, K., Clark, D. A., Hsia, H. C., & Horsley, V. (2018). Myofibroblast proliferation and heterogeneity are supported by macrophages during skin repair. *Science (New York, N.Y.)*, *362*(6417), eaar2971.

Original Comment 2:

While this reviewer acknowledges the authors' effort in performing a non-targeted metabolomic analysis, the rationale for selecting lactate as the central metabolite of interest still remains insufficiently justified. Specifically, it is unclear whether lactate was directly detected and quantified in the presented metabolomic dataset, and if so, whether it represents the most significantly upregulated metabolite in hypertrophic scar (HS) compared with normal skin. If lactate is not the top metabolite showing increased abundance, the authors should clarify why it was prioritized over other metabolites with potentially stronger differential expression. Furthermore, identifying the metabolite(s) with the highest upregulation and discussing their potential relevance would provide critical context, ensuring that the mechanistic focus on lactate is data-driven rather than selectively interpretive.

Reply:

We sincerely appreciate the reviewer's insightful comment. We agree that relying solely on the previous non-targeted metabolomics data was insufficient to fully justify selecting lactate as the primary candidate, especially without a rigorous comparison to other highly upregulated metabolites.

To address this concern and provide a solid data-driven rationale, we have performed a new **Targeted Energy Metabolomics** analysis on HS and NS tissues. Furthermore, we have thoroughly analyzed the "top" metabolites identified in both non-targeted and targeted datasets to exclude confounding factors (such as exogenous drug metabolites) and to pinpoint the bioactive molecules most relevant to the fibrotic pathology.

Our specific responses and revisions are detailed below:

1. Verification of Lactate Elevation via Targeted Metabolomics

In our original non-targeted analysis, lactate was detected but was not the single most statistically significant peak due to the inherent ionization competition and dynamic range

limitations of non-targeted LC-MS/MS for small polar molecules. To overcome this, we conducted **Targeted Energy Metabolomics** (using standard curves for absolute quantification). The results confirmed that lactate is indeed one of the most significantly upregulated metabolites in HS. As shown in the new Figure S1D (Volcano plot) and Figure S1E, lactate ranks as the 3rd highest metabolite in terms of log₂FC among all energy-related metabolites. Figure S1F demonstrates a robust quantitative increase in Lactate, ATP, and Pyruvate. Figure S1G (KEGG enrichment) confirms that "Pyruvate metabolism," "Carbon metabolism," and "Glycolysis" are the enriched pathways, supporting lactate's central role.

2. Rationale for Prioritizing Lactate over the "Top" Upregulated Metabolites

Following the reviewer's suggestion, we scrutinized the metabolites with higher upregulation than lactate to ensure our focus was not selectively interpretive. (1) Analysis of the top metabolite in non-targeted data: The metabolite with the highest fold change in our non-targeted dataset was *N-Desmethyaminopyrine*. Upon rigorous literature review, we identified this as a major metabolite of Aminophenazone/Metamizole, an analgesic and anti-inflammatory drug (L Cardon et al., 1958). Since HS patients frequently undergo surgical excision or receive pain management, the presence of this compound is an exogenous pharmacological artifact rather than an endogenous driver of fibrosis. Therefore, we excluded it from mechanistic investigation. (2) Analysis of top metabolites in targeted data (GTP and DL-Glycerate): In our targeted panel, two metabolites showed a slightly higher log₂FC than lactate: Guanosine Triphosphate (GTP) and DL-Glycerate. GTP is essential for protein synthesis and gluconeogenesis. While elevated GTP supports the high proliferative state of fibroblasts, it is a general indicator of high anabolic activity rather than a specific signaling molecule that triggers the epigenetic shift we observed. Glycerate is an intermediate in serine metabolism and glycolysis. Its accumulation supports the metabolic shift (Warburg effect) but lacks the documented capacity to directly modify histones (Lactylation) or act as an intercellular signaling transmitter (via MCT1) to the same extent as lactate (Sheng Hui et al., 2017). (3) **Why Lactate (Rank 3) is the Key:** Lactate is not merely a fuel source or byproduct; it is a bioactive signaling molecule. Our study focuses on histone lactylation

(H3K231a). Lactate is the **obligate substrate** for this modification (Di Zhang et al., 2019). Furthermore, the concurrent upregulation of MCT1 (lactate transporter) suggests an active transport mechanism specifically for lactate, which is not shared by GTP or Glycerate. Therefore, biologically and mechanistically, lactate is the most relevant driver of the phenotypic remodeling observed.

3. Clarification on Discrepancies between Non-Targeted and Targeted Results

The reviewer may note differences between the two datasets. Non-targeted metabolomics provides a broad "snapshot" but suffers from ion suppression and lower sensitivity for small polar acids like lactate. Targeted metabolomics uses optimized extraction methods and internal standards, providing superior sensitivity and accuracy for energy metabolites (Anton Ribbenstedt et al., 2018). This explains why lactate's significance is more pronounced and reliable in our new targeted dataset.

We have revised the manuscript to reflect these findings: We have inserted the Targeted Metabolomics data (New Figure S1D-G) and explicitly mentioned the exclusion of N-Desmethylaminopyrine as a drug artifact. Moreover, we added a section justifying the selection of lactate over GTP/Glycerate, emphasizing lactate's unique role in epigenetic regulation (lactylation) and cell-to-cell transport (MCT1). We believe these additional experiments and analyses provide the robust justification the reviewer requested. We are grateful for your constructive criticism, which has significantly improved the quality and clarity of our manuscript. Thank you again for helping us strengthen the rigor of our study.

Revised figures:

Figure S1D-G

Revised manuscript:

“KEGG analysis of the different metabolites revealed enrichment in the central carbon metabolism (Figure S1C). To further validate the metabolic profile and overcome the limitations of non-targeted metabolomics in quantifying small polar acids, we performed targeted energy metabolomics which revealed a distinct metabolic signature in HS tissues (Figure S1D). Lactate ranked as the third most upregulated metabolite (log₂FC) following Guanosine Triphosphate (GTP) and DL-Glycerate (Figure S1E). Consistent with a shift toward aerobic glycolysis, quantitative analysis confirmed significantly elevated levels of Lactate, Pyruvic acid, and ATP in HS tissues (Figure S1F). KEGG pathway analysis of these metabolites highlighted enrichment in Pyruvate metabolism, Carbon metabolism, and Glycolysis (Figure S1G).”

“Our combined metabolomic approach clarifies the metabolic landscape of HS. While non-targeted analysis provided a broad overview, targeted energy metabolomics allowed for the precise quantification of glycolytic intermediates. A critical evaluation of the most upregulated metabolites was necessary to identify the pathogenic driver. We excluded N-

Desmethylaminopyrine as a pharmacological artifact. Among endogenous metabolites, while GTP and DL-Glycerate were highly upregulated—reflecting the increased biosynthetic demand of proliferating fibroblasts—we prioritized lactate for mechanistic investigation.”

References:

- (1) CARDON, L., COMESS, O. H., NOBLE, T. A., & POMARANC, M. M. (1958). Value of aminopyrine. *Annals of internal medicine*, *48*(3), 616–634.
- (2) Hui, S., Ghergurovich, J. M., Morscher, R. J., Jang, C., Teng, X., Lu, W., Esparza, L. A., Reya, T., Le Zhan, Yanxiang Guo, J., White, E., & Rabinowitz, J. D. (2017). Glucose feeds the TCA cycle via circulating lactate. *Nature*, *551*(7678), 115–118.
- (3) Zhang, D., Tang, Z., Huang, H., Zhou, G., Cui, C., Weng, Y., Liu, W., Kim, S., Lee, S., Perez-Neut, M., Ding, J., Czyz, D., Hu, R., Ye, Z., He, M., Zheng, Y. G., Shuman, H. A., Dai, L., Ren, B., Roeder, R. G., ... Zhao, Y. (2019). Metabolic regulation of gene expression by histone lactylation. *Nature*, *574*(7779), 575–580.
- (4) Ribbenstedt, A., Ziarrusta, H., & Benskin, J. P. (2018). Development, characterization and comparisons of targeted and non-targeted metabolomics methods. *PloS one*, *13*(11), e0207082.

Original Comment 3:

The addition of transcript-level measurements for selected TCA cycle enzymes is appreciated; however, this approach alone does not constitute definitive evidence of a metabolic shift from oxidative phosphorylation (OXPHOS) to glycolysis in hypertrophic scar (HS) tissue. Given that the authors have already provided Western blot data for glycolysis-related proteins, equivalent protein-level validation for the selected OXPHOS-associated genes would be necessary for methodological consistency and to strengthen this comparison.

Furthermore, the added metabolomic analysis requires deeper exploration. The authors should provide a more comprehensive overview of the metabolites involved in OXPHOS and glycolysis, demonstrating whether OXPHOS intermediates are indeed reduced in HS and whether glycolytic intermediates are correspondingly elevated. Highlighting only a single metabolite (citric acid) from the TCA cycle does not sufficiently support the claim of an energy pathway shift, particularly since other relevant pathways, e.g., fatty acid oxidation, have not been considered.

Finally, expanding the Seahorse data presented in Figure 2 to include an OCR/ECAR ratio would provide a functional readout of the relative balance between oxidative and glycolytic metabolism, thereby offering further support for the proposed shift toward glycolysis.

Reply:

We would like to express our sincere gratitude for your insightful and constructive comments. We agree that protein-level validation and a more comprehensive functional analysis of the metabolic shift are essential to strengthen the conclusions of our study. In response to your suggestions, we have performed additional Western blot assays, conducted a deeper analysis of our metabolomic data (including targeted metabolomics), and re-analyzed our Seahorse data to calculate ATP production rates. Our point-by-point responses are detailed below:

1. Protein-level validation of OXPHOS-associated genes.

We fully agree with the reviewer that Western blot validation is necessary to confirm the

downregulation of OXPHOS at the protein level. In the revised manuscript, we have examined the protein expression of key OXPHOS-associated enzymes, specifically PDHA1, CS, IDH1, and IDH2 in NS and HS tissues. As shown in Figure S1H of the revised manuscript, the results demonstrate a distinct downregulation of these OXPHOS-related proteins in HS tissues compared to NS. Specifically, quantification revealed statistically significant decreases in IDH1 and IDH2 levels. This protein-level data corroborates our RT-qPCR findings and provides robust evidence that the TCA cycle and oxidative phosphorylation capacity are suppressed in HS tissue, supporting the conclusion of a metabolic switch from OXPHOS to glycolysis. We have updated the Results section to incorporate these findings.

2. Deeper exploration of metabolomic analysis and the specific metabolic profile.

We fully appreciate this suggestion. We have expanded our analysis to provide a comprehensive overview of the metabolic landscape in HS. As shown in Figure S1C, KEGG enrichment analysis of our non-targeted metabolomics revealed that, in addition to central carbon metabolism, pathways such as Amino acid biosynthesis, mTOR signaling, PI3K-Akt signaling, and Linoleic acid metabolism were significantly enriched. This suggests that the metabolic reprogramming in HS is a complex network supporting rapid cell proliferation and matrix synthesis, driven by both glycolytic flux and anabolic signaling.

To overcome the limitations of non-targeted metabolomics in quantifying small polar acids, we utilized targeted energy metabolomics. As presented in Figure S1F, we observed significantly elevated levels of Lactate, Pyruvate, and ATP in HS tissues. The simultaneous accumulation of pyruvate and lactate, coupled with the downregulation of TCA enzymes (IDH1/2), strongly supports a "Warburg-like" shift where pyruvate is shunted toward lactate production rather than entering the TCA cycle efficiently. We also noted an elevation in Malate in the HS group (Figure S1E). While a reduction in TCA activity typically lowers TCA intermediates (like the observed decrease in Citric acid), the elevation of Malic acid is consistent with metabolic reprogramming seen in fibroproliferative disorders and cancer. Literature indicates that Malate accumulation often results from: (1) The malate-aspartate

shuttle: To sustain high glycolytic rates, cells must regenerate cytosolic NAD⁺. Malate is a key component of the shuttle system that transports reducing equivalents into the mitochondria, indirectly supporting high cytosolic glycolysis (Jason W Locasale et al., 2011). (2) Anaplerosis (Glutaminolysis): Proliferating cells often utilize glutamine anaplerosis to replenish the TCA cycle. Glutamine-derived carbon can generate Malate, which is then converted to Pyruvate by Malic Enzyme to generate NADPH, which is crucial for lipid and nucleotide synthesis required for HS formation (Ralph J DeBerardinis et al., 2008). Therefore, the increase in Malate does not contradict the suppression of OXPHOS but rather reflects the high biosynthetic and redox demands of the hypertrophic scar microenvironment.

We have revised the Results section to include this deeper interpretation of the metabolomic profile.

3. Functional readout of the metabolic shift (OCR/ECAR vs. ATP Production Rate).

Thank you for this excellent suggestion. To provide a more functional and intuitive readout of the balance between oxidative and glycolytic metabolism, we analyzed the ATP Production Rate derived from our Seahorse data. This metric specifically distinguishes between **ATP produced via Glycolysis (glycoATP)** and **ATP produced via Oxidative Phosphorylation (mitoATP)**, offering a clearer picture than the raw OCR/ECAR ratio. As shown in the new Figure S3A and S3B, macrophages cultured on stiff substrates (50 kPa) or treated with TGF- β exhibited a significant shift in their bioenergetic profile. The contribution of glycolytic ATP production was markedly increased compared to oxidative ATP production in these stimulated macrophages. This confirms that mechanical and inflammatory stimuli drive macrophages toward a glycolytic phenotype. Crucially, as shown in Figure S3C and S3D, neither endothelial cells nor fibroblasts exhibited a comparable increase in the glycolytic ATP ratio under high stiffness conditions. Most importantly, corroborating our cell line data, primary macrophages from HS tissue demonstrated a significantly higher rate of ATP production from glycolysis compared to OXPHOS (Figure S4D). This validates our hypothesis that macrophages are the primary source of the elevated lactate in the HS microenvironment, while fibroblasts are the downstream effectors that utilize this lactate. We

have integrated these ATP production rate analyses into the Results section and updated the relevant figure legends.

We believe these additional experiments and analyses fully address your concerns and significantly strengthen the manuscript. Thank you again for your time and guidance.

Revised figures:

Figure S1H

Figure S1C

Figure S1G

Figure S3

Figure S4D

Revised manuscript:

“KEGG analysis of the different metabolites revealed enrichment in the central carbon metabolism, as well as pathways critical for biosynthesis and proliferation, including amino acid biosynthesis, mTOR signaling, PI3K-Akt signaling, and linoleic acid metabolism (Figure S1C).”

“Moreover, we profiled the expression of these genes in NS and HS using western blotting and the result showed that the protein level of MCT1, MCT4 and PKM2, which were markedly increased in HS, yet IDH1 and IDH2 were significantly decreased (Figure 1C and S1H).”

“To definitively assess the metabolic switch, we calculated the ATP production rates. Macrophages cultured on 50kPa stiff substrates or under TGF- β stimulation exhibited a significant shift where ATP production from glycolysis (glycoATP) markedly exceeded that from oxidative phosphorylation (mitoATP) (Figure S3A and S3B). However, endothelial and fibroblast cultures showed marginal ECAR increases and no significant shift in the glycoATP/mitoATP ratio under stiff conditions (Figure 2E, 2F, S3C and S3D).”

“In addition, enhanced ECAR level and glycoATP/mitoATP ratio was observed in primary

macrophages from HS tissues compared to control (Figure 2H).”

References:

(1) Locasale, J. W., Grassian, A. R., Melman, T., Lyssiotis, C. A., Mattaini, K. R., Bass, A. J., Heffron, G., Metallo, C. M., Muranen, T., Sharfi, H., Sasaki, A. T., Anastasiou, D., Mullarky, E., Vokes, N. I., Sasaki, M., Beroukhim, R., Stephanopoulos, G., Ligon, A. H., Meyerson, M., Richardson, A. L., ... Vander Heiden, M. G. (2011). Phosphoglycerate dehydrogenase diverts glycolytic flux and contributes to oncogenesis. *Nature genetics*, *43*(9), 869–874.

(2) DeBerardinis, R. J., Lum, J. J., Hatzivassiliou, G., & Thompson, C. B. (2008). The biology of cancer: metabolic reprogramming fuels cell growth and proliferation. *Cell metabolism*, *7*(1), 11–20.

Original Comment 7:

While this reviewer acknowledges the substantial effort undertaken to isolate primary cells from patient skin samples and assess their contribution to lactate production, the rationale for excluding other skin-resident cell types remains unclear. Additional cell populations—such as adipocytes, mast cells, and other immune subsets—could plausibly contribute to the altered metabolic landscape of hypertrophic scar. The authors should either extend their analysis to include these dominant skin cell types or provide a clear and experimentally grounded justification for their exclusion. Addressing this point would ensure that the presented conclusions regarding the cellular origin of lactate are comprehensive and not selectively derived.

Reply:

We sincerely thank the reviewer for this critical and constructive feedback. We fully agree that the skin microenvironment is complex and that attributing the metabolic alteration solely to macrophages requires a rigorous exclusion of other potential cellular contributors. To address your concern regarding the "selectivity" of our analysis and to ensure a comprehensive understanding of the lactate source, we have performed additional experimental validations and provided a detailed, grounded justification regarding specific cell populations:

Prompted by your suggestion, we expanded our histological analysis to include Mast cells and T cells, which are significant immune components of the skin. We performed dual immunofluorescence staining on human NS and HS tissues to detect the co-localization of cell-specific markers—Tryptase (for Mast cells) and CD3 (for T cells)—with the key lactate-generating enzymes HK2 and LDHA. As presented in the newly added Figure S4G-S4J, our quantitative analysis revealed that: Unlike the significant upregulation observed in CD68⁺ macrophages, there was no statistically significant difference in the expression levels of HK2 or LDHA within Tryptase⁺ Mast cells or CD3⁺ T cells when comparing HS tissues to NS tissues. This experimental evidence suggests that while these cells are present in the scar

tissue, they do not undergo the profound "glycolytic switch" characteristic of the macrophages in this specific pathological context. Thus, their contribution to the elevated lactate pool appears negligible compared to macrophages.

Regarding adipocytes, we carefully evaluated their potential role based on the specific tissue architecture of hypertrophic scar and established literature. We excluded them as a primary source of lactate for fibroblasts in this study based on two key factors: (1) **Spatial separation:** Histologically, hypertrophic scar is defined by the excessive deposition of extracellular matrix in the reticular dermis, which becomes significantly thickened (Marc G Jeschke et al., 2023). This creates a substantial physical barrier and distance between the active fibrotic core (where fibroblasts are located) and the subcutaneous adipose tissue. Given this spatial segregation, it is unlikely that labile metabolites like lactate secreted by subcutaneous adipocytes would act as the primary paracrine driver for fibroblasts located in the upper and mid-dermis, especially compared to macrophages which are known to physically infiltrate the fibroblast-dense regions. (2) **Mechanistic distinction:** Current literature indicates that the primary contribution of dermal adipocytes to wound healing and scarring is not metabolic secretion, but rather lineage plasticity. Studies have shown that adipocytes promote scarring primarily through the adipocyte-myofibroblast transition, where they transdifferentiate into myofibroblasts (Brett A Shook et al., 2019). Therefore, in the context of our study—which focuses on the *source* of the lactate signal rather than the source of the myofibroblast lineage itself—adipocytes are less likely to be the "producers" of the lactate gradient.

By combining our original data on macrophages/endothelial cells/fibroblasts with these new findings on Mast cells/T cells and the spatial logic regarding adipocytes, we believe we have now provided a comprehensive identification of the cellular origin of lactate. Our data strongly supports the conclusion that macrophages are the dominant driver of the lactate-rich metabolic microenvironment in hypertrophic scar. We have updated the Results section and the Discussion section of the revised manuscript to reflect these new data and justifications. We believe these additional experiments significantly strengthen the mechanistic depth of our

study. Thank you again for your time and guidance.

Revised figures:

Figure S4G

Figure S4H

Figure S4I

Figure S4J

Revised manuscript:

“To further rule out the contribution of other immune cell populations to the lactate-rich microenvironment, we also performed immunofluorescence co-staining of key glycolytic enzymes (HK2 or LDHA) with markers for mast cells (Tryptase) and T cells (CD3) in human NS and HS tissues. Quantitative analysis revealed no statistically significant differences in the number of glycolytic-enzyme-positive mast cells or T cells (Figure S4G-S4J).”

“Although the skin comprises a heterogeneous population of cells, our systematic analysis identifies macrophages as the principal source of lactate.”

“While adipocytes are key components of the skin stroma, their contribution to the lactate

pool in HS was considered less prominent due to spatial constraints; the significantly thickened reticular dermis in HS creates a physical barrier that separates the fibrotic core from the subcutaneous adipose layer, limiting the potential for adipocyte-derived paracrine metabolic signaling to reach dermal fibroblasts. Additionally, our supplementary data confirmed that vascular endothelial cells and other resident immune cells, such as mast cells and T cells, do not exhibit the robust glycolytic upregulation observed in macrophages.”

References:

- (1) Jeschke, M. G., Wood, F. M., Middelkoop, E., Bayat, A., Teot, L., Ogawa, R., & Gauglitz, G. G. (2023). Scars. *Nature reviews. Disease primers*, 9(1), 64.
- (2) Shook, B. A., Wasko, R. R., Rivera-Gonzalez, G. C., Salazar-Gatzimas, E., López-Giráldez, F., Dash, B. C., Muñoz-Rojas, A. R., Aultman, K. D., Zwick, R. K., Lei, V., Arbiser, J. L., Miller-Jensen, K., Clark, D. A., Hsia, H. C., & Horsley, V. (2018). Myofibroblast proliferation and heterogeneity are supported by macrophages during skin repair. *Science (New York, N.Y.)*, 362(6417), eaar2971.

Original Comment 8:

Without experimental verification, the molecular modeling remains speculative. The authors should indicate whether the model's predictions can be experimentally tested. Validation through targeted mutagenesis or binding assays would significantly enhance the credibility of these computational results.

Reply:

We sincerely thank the reviewer for this insightful and critical comment. We agree that while molecular docking provides a structural hypothesis, experimental validation is essential to confirm the specific interaction interface between COL11A1 and MCT1. To address this concern and move beyond speculation, we have performed additional experiments including **reverse Co-Immunoprecipitation** and **site-directed mutagenesis assays** to physically validate the predictions made by our model.

In our original manuscript, we demonstrated the interaction primarily by immunoprecipitating COL11A1. To further confirm the specificity and robustness of this complex, we performed a reverse Co-IP using an anti-MCT1 antibody. The results unequivocally showed that MCT1 pulls down COL11A1 in HSFs, confirming a stable physical interaction between these two proteins in the cellular context.

Based on the molecular docking results presented in Figure 8E of our original manuscript, we identified specific amino acid residues at the binding interface. To test whether these sites are indeed responsible for the interaction, we constructed expression vectors with specific point mutations at these predicted sites. We generated Flag-tagged COL11A1 mutants (FLAG-COL11A1-MUT) and His-tagged MCT1 mutants (His-MCT1-MUT). We then performed Co-IP assays in cells co-transfected with these vectors. The results demonstrated that the interaction was strong between COL11A1-WT and MCT1-WT. Crucially, the interaction was **significantly impeded** when mutations were introduced into the predicted binding interface of either COL11A1 or MCT1. The disruption was most pronounced when both

binding sites were mutated.

These experimental findings provide direct biological evidence supporting our computational model. They confirm that COL11A1 interacts with MCT1 through the specific domains identified in our docking simulation, thereby validating the proposed mechanism by which COL11A1 may stabilize MCT1 and facilitate lactate transport. We have integrated these new data into Figure 8 of the revised manuscript and updated the Results section to reflect this experimental verification. We believe these additional experiments significantly strengthen the mechanistic depth of our study. Thank you again for your time and guidance.

Revised figures:

Figure 8C

Figure 8G

Revised manuscript:

“To confirm this interaction, co-immunoprecipitation (Co-IP) experiments revealed a significant interaction between COL11A1 and MCT1 (Figure 8B and 8C).”

“We then mutated the predicted binding sites between COL11A1 and MCT1 (Figure 8G). Co-IP assays suggested that the interaction between COL11A1 and MCT1 was effectively impeded by mutations in the TRIM34-binding site (FLAG-COL11A1-MUT) or UPF1-binding site (His-MCT1-MUT), whether introduced individually or in conjunction (Figure 8G).”

Response to Reviewers

Dear Editors and Reviewers,

Thank you for your letter and for the reviewers' comments concerning our manuscript entitled "Lactate derived from macrophages drives skin dermal fibroblasts phenotypic remodeling via MCT1-primed histone H3 lysine 23 lactylation in hypertrophic scar" (Manuscript Number: NCOMMS-25-39200). This comment is valuable and very helpful for revising and improving our manuscript, as well as the important guiding significance to our researches. The responses to the reviewer's comments are as following:

Responses to the reviewer's comment:

Reviewer #3 (Remarks to the Author):

The manuscript has been substantially improved. Although this reviewer still has some reservations regarding the rationale and justification for selecting lactate as the key metabolite, the overall quality of the work warrants publication in Nature Communications.

Reply:

We are deeply grateful for your positive assessment of our work and your recommendation for its publication in Nature Communications. We strictly value your reservation regarding the rationale for selecting lactate as the key metabolite, as this is indeed the foundational premise of our study. We would like to take this opportunity to further clarify our screening logic and justification, which combines metabolomic screening with mechanistic specificity:

As presented in our "Targeted energy metabolomic" analysis (Figure S1E and Lines 153-158 of the manuscript), lactate was identified as the third most upregulated metabolite (based on log₂FC) in hypertrophic scar (HS) tissues. The top candidate, N-Desmethylaminopyrine, was excluded as it is a pharmacological artifact (Line 566). The other top candidates, GTP (Guanosine Triphosphate) and DL-Glycerate, were indeed highly upregulated. However, as discussed in our manuscript (Lines 567-568), these metabolites largely reflect the heightened

"biosynthetic demand" and general proliferative state of fibroblasts, which are common features in many proliferative disorders. We prioritized lactate over GTP and DL-Glycerate because our core objective was to identify a metabolite that functions not just as a fuel, but as a signaling molecule capable of driving the specific "phenotypic remodeling" in HS.

Epigenetic Link: Unlike GTP or Glycerate, lactate serves as the specific substrate for histone lactylation (specifically H3K231a, as shown in Figure 4). This aligns with the "Warburg effect" observed in HS (Lines 589-593), offering a direct mechanistic link between metabolic reprogramming and epigenetic gene regulation (the MCT1-H3K231a-HEY2/COL11A1 axis). Our transcriptomic and protein analysis (Figure 1B-C) revealed a significant upregulation of MCT1 (a specific lactate transporter) in HS myofibroblasts, further pinpointing lactate transport as a critical regulatory node in this pathology. Our functional assays confirmed that exogenous lactate (10 mM) was sufficient to induce the expression of key fibrotic markers (COL1A1, α -SMA) in a dose-dependent manner (Figure S2A-B), verifying its role as a "pathogenic driver" rather than a passive bystander.

In summary, while other metabolites were upregulated, lactate was selected because it uniquely bridges the gap between the metabolic microenvironment (macrophage-derived glycolysis) and the epigenetic reprogramming of fibroblasts (histone lactylation), which constitutes the novel mechanism we aimed to elucidate. We hope this explanation effectively addresses your reservation and reinforces the rationale behind our experimental design.